# Bidirectional Collaborative Medical Report Generation via Concept-level Interaction

## Abstract

We introduce the first bidirectional collaborative medical report generation framework to reduce physicians' workload and enhance trustworthiness through targeted physician-AI interaction, where physicians provide feedback only on the most critical parts, and the Vision-Language Model (VLM) propagates these to finalize the full report. The core challenge lies in defining the optimal unit of interaction. We propose the Anatomy-Finding Concept Unit (AFCU), a minimal, clinically grounded semantic statement (e.g., "left lobe: hypoechoic nodule"), satisfying three key principles: atomicity, lack of ambiguity, and anatomical anchoring. To extract AFCU, we use a Large Language Model (LLM) guided by predefined clinical templates followed by information bottleneck clustering to group lexically diverse but semantically equivalent anatomical concepts (e.g., "left and right lobe" to "both lobes of the thyroid gland"), eliminating redundancy while preserving diagnostic fidelity. To prioritize physician intervention, we introduce the Concept Risk Score (CRS), quantifying behavioral inconsistency (concepts generated regardless of image content) and semantic instability (inconsistent associated findings under image perturbations) via occlusion-based visual grounding. Finally, we propose Holistic Semantic Match (HSM), a concept-based metric that correlates strongly with human judgment (Pearson's r = 0.846, $p < 0.05$). Experiments show our framework improves semantic quality by 9.13% HSM across four organs by correcting only one AFCU with high error risk per report – a minimal, clinically feasible intervention, enabling efficient and trustworthy physician-AI collaboration.

## 1 Introduction

Medical imaging reports serve as critical objective evidence for clinical diagnosis but impose substantial time and workload burdens on physicians (Kisilev et al., 2015; Hartsock & Rasool, 2024). Advances in Artificial Intelligence (AI) have significantly improved the accuracy of automated Medical Report Generation (MRG), making it one of the most promising solutions (Li et al., 2021; Zhou, 2023; Wang et al., 2025). However, does automated MRG truly reduce the workload of physicians? In practice, due to the lack of trust stemming from the "black-box" nature of AI systems (Messina et al., 2022), their potential for errors and the associated ethical concerns, physicians still need to conduct comprehensive manual reviews of generated reports. As a result, the final report quality remains heavily dependent on the physician's vigilance and expertise. Since physician involvement is unavoidable, the real challenge lies not in removing physicians from the loop by pushing model accuracy ever higher, which has an upper bound, but in leveraging their expertise more effectively. This motivates a new **collaborative MRG paradigm**, where physicians and AI interact in a targeted and trustworthy manner to jointly produce reliable medical reports.

While the need for effective collaboration between AI and physicians is clear, existing approaches remain largely unidirectional. To structure these approaches and advance collaborative MRG, we define the **Report Collaboration Level** (RCL) based on AI trustworthiness and physicians' actual workload, as illustrated in Figure 1. Current mainstream methods fall into the first two levels. RCL-1 Passive Collaboration. Exemplified by Flamingo-CXR (Tanno et al., 2025), where AI drafts the report and physicians comprehensively revise it, akin to "an intern writing a draft for an attending physician to rewrite." This mode features low AI trustworthiness and high physician burden. RCL-2 Guided Collaboration. As in Keyword-based MRG (Dong et al., 2025), where physicians provide

keywords and AI structures the final report, analogous to "a supervisor providing key points for an intern to compose." Here, AI exhibits partial interpretability, moderately reducing physician workload. However, both RCL-1 and RCL-2 represent unidirectional workflows, underutilizing AI's deep understanding of medical images and text, and still requiring comprehensive manual review.

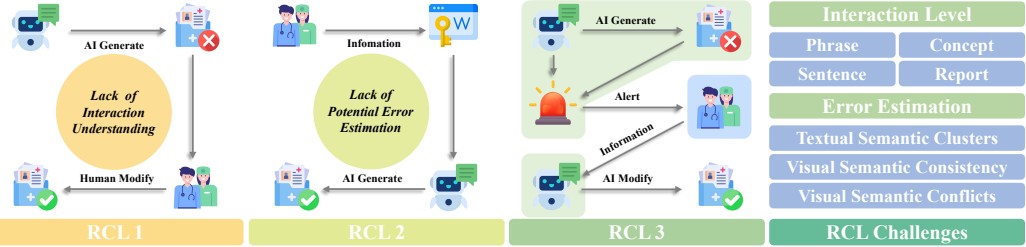

Figure 1: Comparison and Visualization of Report Collaboration Levels (RCL). RCL categorizes human-AI report generation, from full physician review (Tanno et al., 2025; Dong et al., 2025) to interaction focused only on key uncertainties.

To address this limitation, we introduce **RCL-3** — an active bidirectional paradigm for the first time: VLMs draft reports, proactively flag contents with high error risk, and physicians correct only those. VLMs then finalize the report. Analogous to "an intern flags uncertainties for targeted guidance," this boosts AI trustworthiness, reduces physician workload, and enhances collaboration efficiency. However, in RCL-3, a core challenge emerges: ***what is the efficient, yet clinically meaningful, unit a physician should correct?*** Inspired by software engineering — where engineers fix statements, not entire files or tokens — we argue that physicians require an interaction unit that is atomic, unambiguous, and strongly anchored to clinical reality. Figure 2 illustrates that coarse units (e.g., sentences) violate atomicity, forcing review of multi-fact statements; fragmented ones (e.g., GPT concepts) violate unambiguity, being weakly anchored to anatomy (More details can be found in Sec. A.19). We propose the Anatomy-Finding Concept Unit (AFCU) as the optimal solution: a minimal semantic "statement" composed of an anatomical concept (e.g., "left lobe") and its associated finding concepts (e.g., "hypoechoic nodule"). Our empirical analysis in Table 1 demonstrates that AFCU is the optimal choice for achieving the best performance in practice. However, this raises two issues: ***(i) how to extract these clinically grounded concept units?*** and ***(ii) how to assess potential errors?*** Regarding extraction methods, we use DeepSeek-V3 (DeepSeek-AI, 2024) to initially extract anatomical and finding concepts based on predefined templates derived from real clinical reports, followed by clustering anatomical concepts to reduce semantic redundancy using information bottleneck. For potential error assessment, we propose the Concept Risk Score (CRS). CRS quantifies each anatomical concept's visual detachment and semantic instability through occlusion-based perturbations and automatically identifies high-risk anatomical concept requiring physician intervention. A higher CRS indicates greater model improvement. Physicians then provide feedback only on one flagged anatomical concept, enabling VLMs to regenerate more reliable reports based on concept-level interaction.

Moreover, better medical report generation must be evaluated with clinically grounded metrics. BLEU (Papineni et al., 2002) and ROUGE (Lin, 2004) capture only surface text; even HalFscore (Chen et al., 2025a) — reliant on LLMs and natural image priors — is impractical for structured reports. We propose a concept-based metric, Holistic Semantic Match (HSM), that measures entity coverage, attribute fidelity, and clinical alignment. As shown in Figure 5(a), our proposed metric exhibits strong correlation with human evaluations (Pearson's r = 0.846, $p < 0.05$), validating its clinical effectiveness.

To summarize, our main contributions are threefold. First, we introduce the first bidirectional collaborative MRG framework, which improves model trustworthiness while reducing physicians' workload. Second, via information bottleneck theory, we enable VLMs to interpret and revise concept-level feedback through clustering and concept-level instruction tuning for targeted report refinement. Third, we propose the Concept Risk Score, a perturbation-based metric that prioritizes high-leverage anatomical entity corrections to maximize diagnostic gain per intervention. Additionally, we establish the first concept-grounded semantic metric suite for medical reports, and by flagging the single most critical entity concept for intervention, our method boosts HSM semantic similarity by an average of 9.13% across four organs, achieving significant quality gains with minimal physician intervention.

## 2 RELATED WORK

### 2.1 MEDICAL REPORT GENERATION

Medical report generation is a cornerstone of computer-aided diagnosis, aiming to alleviate clinicians' workload (Liu et al., 2025b; Chen et al., 2025b). Deep learning has driven steady progress (Jing et al., 2017; Zhang et al., 2017; Zeng et al., 2020; Jin et al., 2024b; Tang et al., 2025) — from early CNN-LSTM hybrids SAT (Vinyals et al., 2015), to memory-augmented Transformers R2Gen (Chen et al., 2020), and knowledge-bridged architectures KMVE (Li et al., 2024a). More recently, Vision-Language Models have further improved fluency and coverage (Hartsock & Rasool, 2024; Ge et al., 2025). Concurrently, other efforts have sought to enhance factual consistency through knowledge graphs (Zhang et al., 2020; Li et al., 2023; Hou et al., 2023a), tree-structured observation planning (Hou et al., 2023b), or clinical knowledge injection into Transformers (Huang et al., 2023). While these approaches enhance report quality, anatomical grounding, and structured reasoning, they remain fully automatic and operate within a unidirectional generation paradigm. Consequently, their outputs still require comprehensive review and correction by physicians in clinical practice. Due to limited trust and lack of interactive refinement mechanisms, this paradigm (Tanno et al., 2025; Dong et al., 2025) has not significantly reduced physician workload, highlighting the urgent need for a bidirectional collaboration framework that enables concept-level interaction and shared goal understanding.

### 2.2 HUMAN-AI COLLABORATION

Most human activities are collaborative, so integrating AI into complex workflows requires a Computer-Supported Cooperative Work perspective (Wang et al., 2020). Human-AI collaboration has reduced human workload across various domains. In systematic literature reviews (Spillias et al., 2024), AI-assisted retrieval and screening enhanced accuracy, achieving low omission rates and high consistency despite some false positives. In brain MRI differential diagnosis (Kim et al., 2025), radiology residents using LLM-assisted search tools improved diagnostic accuracy without affecting interpretation time or confidence. However, current approaches in medical report generation (Tanida et al., 2023) largely involve passive, unidirectional collaboration, such as Flamingo-CXR (Tanno et al., 2025) reports for physicians to revise or physicians providing keywords for AI to organize (Dong et al., 2025). These methods don't significantly reduce clinicians' workload.

### 2.3 UNCERTAINTY ESTIMATION FOR LARGE MULTIMODAL MODELS

As multimodal large models spread, assessing output reliability and using uncertainty estimation to identify potential errors has become a key challenge Liu et al. (2025a). Current uncertainty estimation methods can be broadly categorized into sequence-level and entity-level approaches. Sequence-level methods, such as token probability-based uncertainty (Guerreiro et al., 2022) and semantic entropy via clustering (Kuhn et al., 2023; Farquhar et al., 2024), capture global output variability but fail to localize errors to specific entities. VL-Uncertainty (Zhang et al., 2024) improves robustness via visual-textual perturbations but remains sequence-focused. Although recent entity-level detection methods (Obeso et al., 2025) attempt fine-grained validation, their reliance on external knowledge bases makes it difficult to meet the core requirement of visual-grounded accuracy in medical imaging reports — even when combined with Retrieval-Augmented Generation, they cannot ensure consistency between the generated content and the visual features of the image (e.g., echo, boundaries, blood flow), leading to clinical risk. Furthermore, existing methods fail to provide intervention priorities based on entity-level uncertainty.

## 3 METHODOLOGY

We propose a concept-centric, risk-aware framework for human-AI collaborative medical report generation, addressing three key challenges: extracting non-redundant concepts, integrating human feedback without retraining, and prioritizing high-impact interventions. Our pipeline begins with Compression of Anatomical Concepts via Information Bottleneck (§3.1), which distills reports into a structured, image-grounded concept dictionary, compressing redundancy while preserving critical Anatomy-Finding Concept Units (AFCU). Next, Concept Instruction Tuning enables models to

self-calibrate during inference (§3.2), aligning generated reports with physician-provided concept cues to reflect high-confidence clinical knowledge when uncertainty arises. We then introduce the Concept Risk Score (§3.3), a two-stage metric that identifies concepts most likely to benefit from intervention, evaluating behavioral inconsistency and semantic ambiguity to highlight optimal targets for correction. Finally, the Holistic Semantic Match metric assesses clinical fidelity by measuring semantic alignment (§3.4), focusing on anatomical accuracy and descriptive consistency rather than lexical overlap.

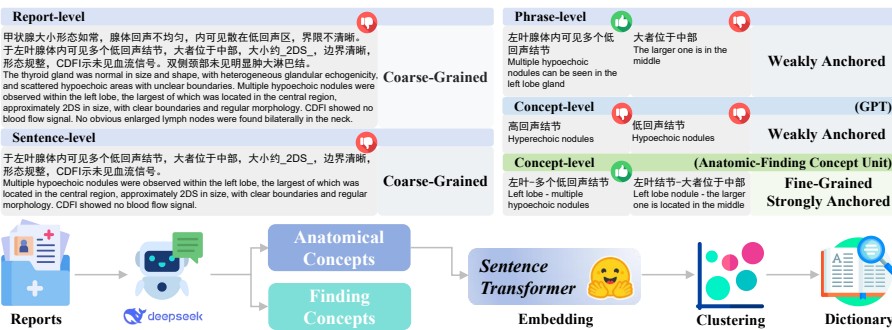

Figure 2: Comparison of Interaction Levels by Semantic Granularity and Anchoring, with Anatomic-Finding Concept Unit Extraction Pipeline. Details of the different granularity levels are provided in Sec. A.19.

Realistic human-AI collaborative report generation follows the three-stage pipeline in Figure 3: (1) A fine-tuned VLM generates an initial report; (2) CRS identifies the anatomical concept with the highest error risk, and a physician provides the corrected description; (3) The VLM incorporates this feedback to produce the final revised report.

## 3.1 COMPRESSION OF ANATOMICAL CONCEPTS VIA INFORMATION BOTTLENECK

Recall from Section 1 that we define the **Anatomy-Finding Concept Unit (AFCU)** as the atomic unit of human-AI collaboration – composed of an *anatomical concept* (e.g., "left lobe") and its associated *finding concepts* (e.g., "hypoechoic nodules"). While finding concepts are clinically discriminative and must remain uncompressed, anatomical concepts suffer from lexical redundancy. For example, "bilateral thyroid lobes", "left and right lobe", and "bilateral glandular tissue" all describe the same anatomy. This variation hinders efficient interaction. As shown in Figure 4(a), compressing only anatomical concepts reduces redundancy by **87.1%**, while preserving diagnostic fidelity through uncompressed findings.

Given image $X$ and report $R$, our goal is to extract a structured, non-redundant concept dictionary $\mathcal{D}$ that preserves maximal semantic relevance to $X$ and eliminates linguistic redundancy in $R$. This is formalized as an Information Bottleneck (IB) (Tishby et al., 2000; Tishby & Zaslavsky, 2015) objective:

$$\mathcal{E}^* = \arg\max_{\mathcal{E}' \subseteq \mathcal{E}} \left[ I(\mathcal{E}'; X) - \beta \cdot I(\mathcal{E}'; \mathcal{E}) \right], \tag{1}$$

where $\mathcal{E}$ are extracted anatomical concepts, $\mathcal{E}^*$ is the compressed version, and $\beta$ balances relevance versus conciseness.

Since mutual information is intractable to compute directly, we approximate Eq. 1 in three steps.

Firstly, extract high-fidelity concepts. Use DeepSeek-V3 (DeepSeek-AI, 2024) and clinical templates (see Appendix § A.3) to extract Anatomical Concept set $\mathcal{E}$ (e.g., "left and right lobe", "bilobar lobe") and Finding Concept set $\mathcal{A}$ (e.g., "size normal", "uniform echo"). Since reports describe $X$, extracted anatomical concepts already exhibit high $I(\mathcal{E}; X)$ — we start from a high-fidelity subspace. Secondly, cluster semantically equivalent anatomical concepts. Encode each $e_i \in \mathcal{E}$ into $\phi(e_i) \in \mathbb{R}^d$ via Sentence Transformer (Reimers & Gurevych, 2019). Anatomical concepts are clustered if their cosine similarity exceeds an adaptive threshold $\mu_s + \gamma \cdot \sigma_s$, where $\mu_s$ and $\sigma_s$ are the mean and standard deviation of all pairwise similarities. The compression strength parameter $\gamma > 0$

is set by clinical experts based on desired granularity — higher $\gamma$ yields more, narrower clusters. The resulting canonical entity set $\mathcal{E}^*$ is formally defined as:

$$\mathcal{E}^* = \left\{ e_k \,|\, \exists \mathcal{C}_k \subseteq \mathcal{E} \text{ s.t. } \forall e_i, e_j \in \mathcal{C}_k, \, s(e_i, e_j) > \mu_s + \gamma \cdot \sigma_s \right\}. \tag{2}$$

For each cluster $\mathcal{C}_k$, a clinical expert selects the most appropriate representative $e_k^* \in \mathcal{C}_k$ based on clinical canonical usage and report clarity, ensuring $\mathcal{E}^*$ remains clinically faithful. When $s(e_i, e_j)$ is high, $e_i$ and $e_j$ are semantically equivalent, satisfying $I(e_i; e_j) \approx H(e_i)$. Merging them reduces $I(\mathcal{E}^*; \mathcal{E})$ while preserving $I(\mathcal{E}^*; X)$. Thirdly, attach findings without compression. For each $e_k^*$, retain all associated finding concepts $\mathcal{A}_{e_k^*} \subseteq \mathcal{A}$, forming $\mathcal{D} = \left\{ (e_k^*, \mathcal{A}_{e_k^*}) \right\}_{k=1}^{K}$. Finding concepts (e.g., "hypoechoic", "irregular margin") are diagnostic modifiers — compressing them risks critical loss. Thus, we compress only anatomical concepts, not finding concepts. As shown in Figure 2, our extract, cluster, and bind pipeline approximates the IB objective, yielding a compact, clinically faithful concept dictionary for RCL-3's concept instruction tuning, risk scoring, and semantic evaluation (see Appendix § A.3 for more details).

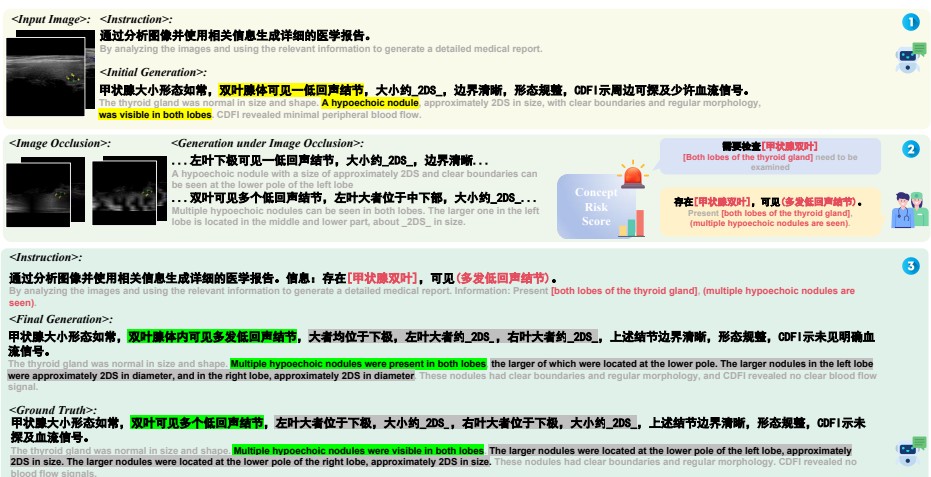

Figure 3: The three main stages of bidirectional human-AI collaborative report generation. Among them, Concept Instruction Tuning follows the same form as the third stage.

## 3.2 CONCEPT INSTRUCTION TUNING

We implement anatomically grounded guidance via concept instruction tuning : during training, we fine-tune the vision encoder, projector and LLM via LoRA (Hu et al., 2022) (see Appendix § A.6) on triplets $(I, x_q, x_p)$, where $I$ is the input image; $x_q$ is the base instruction ("generate a detailed report"); and $x_p$ is a minimal physician cue in AFCU format — e.g., ``Present [both lobes of the thyroid gland], (multiple hypoechoic nodules are seen).'' — injected as contextual instruction as illustrated in Figure 3 Part 3, where *"both lobes of the thyroid gland"* denotes an anatomical concept and *"multiple hypoechoic nodules are seen"* a finding concept. Crucially, during inference, no retraining is needed: physicians can provide $x_p$ in the same format to guide generation. The model, having learned to reconcile $x_p$ with visual input during SFT, performs self-calibration — using $x_p$ as a high-confidence

Table 1: Average performance of interventions at different levels after instruction Supervised Fine-Tuning (SFT). Concept-level (AFCU) SFT refers to our proposed Concept Instruction Tuning. Implementation details are provided in Sec. A.19.

| Type | BLEU-4 | ROUGE-L | HSM |
| --- | --- | --- | --- |
| SFT | 0.6341 | 0.7277 | 0.6475 |
| + Phrase-level | 0.5662 | 0.6775 | 0.5984 |
| + Sentence-level | 0.6481 | 0.7380 | 0.6280 |
| + Report-level | **0.8669** | **0.8999** | **0.8681** |
| Concept-level (GPT) SFT | 0.5653 | 0.6434 | 0.5602 |
| + Concept-level (GPT) | 0.5131 | 0.6370 | 0.5470 |
| Concept-level (AFCU) SFT | 0.6230 | 0.6985 | 0.6282 |
| + Concept-level (AFCU) | 0.6604 | 0.7413 | 0.7015 |

semantic anchor to redirect attention toward clinically critical features. Figure 3 Part 3 demonstrates this: from the minimal cue above, the model generates new, clinically accurate details absent in $x_p$ — e.g., "larger nodules at the lower pole, 2DS in diameter" (grey text). See Appendix § A.4 for training examples.

## 3.3 CONCEPT RISK SCORE

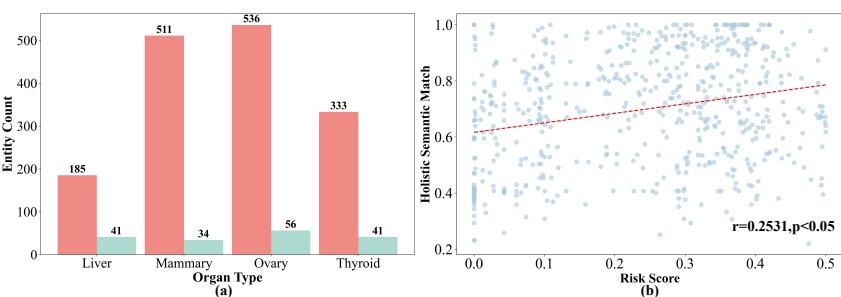

Figure 4: (a) The number of entities in free-text medical reports before and after extraction using Non-redundant Concepts. (b) The correlation between Concept Risk Score and post-intervention performance.

In clinical VLM-assisted radiology workflows, manual verification of all generated reports is prohibitively time-consuming for physicians. To guide efficient physician feedback, we propose the Concept Risk Score (CRS) that identifies which anatomical concepts are most likely to be visually ungrounded by jointly detecting two failure patterns: (1) the model generates the anatomical concept too consistently across image perturbations (behavioral rigidity), indicating it ignores visual evidence; and (2) when the anatomical concept appears, its associated finding concepts (e.g., size, margin, echogenicity) vary semantically across perturbations (content uncertainty), indicating unstable visual grounding. CRS multiplies these signals so that only anatomical concepts that are both persistently generated and semantically inconsistent in their findings receive high scores — precisely those where physician correction will most improve vision-language alignment. In practice, this means CRS automatically surfaces the highest-leverage errors: fix one anatomical concept's description, and the model's behavior improves disproportionately.

CRS is computed using the outputs from 7 independent random grid occlusions ($T_1, \ldots, T_7$) and the original unoccluded image ($T_0$) (see Appendix § A.7). For each standardized anatomical concept $e$, we evaluate behavioral rigidity and content uncertainty.

**Behavioral Rigidity** — whether $e$ is generated too consistently across the 7 occluded outputs. We compute frequency as Freq($e$) equal to the number of outputs containing $e$ divided by 7. Stability is derived from the binary entropy of $e$'s appearance pattern: $H(e) = -\sum_{x \in \{0,1\}} P(x) \log_2 P(x)$, where $P(X = 1) = $ Freq($e$). Stability is normalized to [0,1] as $1 - H(e)$, peaking when $e$ appears always or never. The product Freq($e$) × Stability($e$) captures "false robustness" — anatomical concepts generated regardless of image content.

**Content Uncertainty** — whether the finding concepts associated with $e$ (e.g., "irregular margin") fluctuate semantically across occlusions. We compute the Semantic Ambiguity Index (SAI) as SAI($e$) $= \sqrt{|s(1-s)|}$, where $s$ is the average cosine similarity using Sentence Transformer (Reimers & Gurevych, 2019) between finding concept phrases in $T_1, \ldots, T_7$ and those in $T_0$. SAI peaks at $s = 0.5$, highlighting cases where findings are neither preserved nor random — maximally ambiguous.

The final score is:

$$\text{CRS}(e) = \text{Freq}(e) \times \text{Stability}(e) \times \text{SAI}(e). \tag{3}$$

High CRS indicates an anatomical concept that is frequently and stably generated (visually disengaged) yet accompanied by inconsistent finding concepts (visually ambiguous). Example: "thyroid nodule" always appears, but its findings jump from "irregular margin" to "smooth margin" — a prime candidate for correction.

The theoretical foundation of this design stems from a core principle of visual grounding: a model that genuinely generates reports based on visual evidence should be both input-sensitive and input-consistent. Specifically: (1) If the model truly relies on visual evidence, it should stop reporting an anatomical concept (e.g., "nodule") when its region is occluded ($T_1, \ldots, T_7$). Persistent generation (high frequency/low entropy) means the model ignores visual input—like a robot that always

says "I see a dog" even when the picture is covered. (2) If the model does report a concept, its description (e.g., "smooth margin") should stay stable across views. Wild fluctuations (high Semantic Ambiguity Index, SAI) mean unstable understanding—like describing the same dog as "brown" one time and "black" the next. CRS multiplies these signals to flag only concepts that are both visually disengaged (shouldn't appear) and semantically ambiguous (described inconsistently)—exactly the high-risk errors requiring physician correction.

### 3.4 HOLISTIC SEMANTIC MATCH METRIC

Clinical report generation requires precision, not just fluency. BLEU (Papineni et al., 2002) and ROUGE (Lin, 2004) ignore anatomy-finding semantics. BertScore (Zhang et al., 2019) leverages contextual embeddings but models the report holistically, lacking fine-grained alignment between anatomical and finding concepts. Evaluation must assess correct anatomical concepts and accurate finding concepts. To address this, we propose Holistic Semantic Match (HSM) — a clinically grounded metric that evaluates two essential dimensions: (1) correct identification of anatomical concepts, and (2) semantic accuracy of their associated finding concepts. HSM combines both via geometric mean, forcing models to excel at both — no trade-offs allowed.

Given generated report $R_{\text{pred}}$ and ground truth $R_{\text{gt}}$, we normalize anatomical concept surface forms using the canonical dictionary $\mathcal{D}$ introduced in Section 3.1 (e.g., "right lobe of liver" to "right lobe"), then extract sets $E_{\text{pred}}$ and $E_{\text{gt}}$. Coverage is measured by Anatomical Intersection over Union (AIOU):

$$\text{AIOU} = \frac{|E_{\text{pred}} \cap E_{\text{gt}}|}{|E_{\text{pred}} \cup E_{\text{gt}}|}. \tag{4}$$

For each anatomical concept $e \in E_{\text{pred}} \cup E_{\text{gt}}$, we extract its finding concepts $A_{\text{pred}}(e)$ and $A_{\text{gt}}(e)$ (empty if missing), and compute semantic similarity using Sentence Transformer (Reimers & Gurevych, 2019): $\text{sim}(e) = \cos\left(\phi(A_{\text{pred}}(e)), \phi(A_{\text{gt}}(e))\right)$, averaged over concepts with at least one non-empty finding, yielding the Finding Semantic Similarity (FSS):

$$\text{FSS} = \frac{1}{|\mathcal{E}|} \sum_{e \in \mathcal{E}} \text{sim}(e), \quad \text{where} \quad \mathcal{E} = \{e \mid A_{\text{pred}}(e) \neq \emptyset \text{ or } A_{\text{gt}}(e) \neq \emptyset\}. \tag{5}$$

Final score:
$$\text{HSM} = \sqrt{\text{AIoU} \times \text{FSS}}. \tag{6}$$

HSM is interpretable: low AIoU indicates missed or hallucinated anatomy; low FSS indicates inaccurate descriptions. Only when both are high does HSM reward the output — aligning evaluation with clinical safety. A complete HSM computation example is provided in Appendix A.5. For comparisons with other Sentence Transformers, see Appendix A.8.

## 4 EXPERIMENTS

### 4.1 COMPARATIVE METHODS AND IMPLEMENTATION DETAILS

We selected four existing approaches for comparison: (1) SAT (Vinyals et al., 2015), based on CNN and hierarchical LSTM; (2) R2Gen (Chen et al., 2020), which incorporates a memory-driven unit into the Transformer; (3) KMVE (Li et al., 2024a), an unsupervised prior knowledge-guided method; and (4) fine-tuned Qwen2.5-VL (Bai et al., 2025) 3B and 7B models (Ge et al., 2025). To ensure reliability and credibility, we evaluated all methods on the open-source USReport dataset (Li et al., 2024b) (covering Thyroid, Mammary, and Liver) and a private multi-cancer ovarian ultrasound report dataset (see Appendix §A.2). All data and experiments are in Chinese. English text in figures was translated from Chinese using Google Translate for readability. We fine-tuned the vision encoder, projector, and LLM with LoRA(Hu et al., 2022) in Qwen2.5-VL 3B/7B as our primary experimental models (additional VLM experiments in Appendix §A.10). All hyperparameters and implementation details are provided in Appendix §A.6 for reproducibility.

## 4.2 MAIN RESULTS

Table 2: Results of our method and baselines. All VLM experiments are based on Qwen2.5-VL and incorporate either SFT (Ge et al., 2025) or our Concept Instruction Tuning and Intervention. Top-1 and top-2 results are highlighted in best and second, respectively. UB denotes the theoretical upper bound under Concept Instruction Fine-tuning with intervention, specifically defined as the result obtained by inputting the complete ground-truth report as a prompt into the VLM after supervised fine-tuning.

| Datasets | Methods | NLG METRICS | | | | CE METRICS | | | SEMANTIC METRICS | | |
|---|---|---|---|---|---|---|---|---|---|---|---|
| | | BLEU-1 | BLEU-4 | METEOR | ROUGE-L | Precision | Recall | F1 Score | AIOU | FSS | HSM |
| Thyroid | SAT | 0.1127 | 0.0825 | 0.1502 | 0.3533 | 0.8083 | 0.3895 | 0.5110 | 0.3614 | 0.1880 | 0.2512 |
| | R2Gen | 0.6053 | 0.4735 | 0.3557 | 0.6688 | 0.8678 | 0.7342 | 0.7847 | 0.6656 | 0.3862 | 0.4997 |
| | KMVE | 0.7256 | 0.6113 | 0.4058 | 0.7085 | 0.8304 | 0.8638 | 0.8307 | 0.7368 | 0.5179 | 0.6101 |
| | 3B SFT | 0.7532 | 0.6341 | 0.4226 | 0.7277 | 0.8509 | 0.8910 | 0.8596 | 0.7752 | 0.5531 | 0.6475 |
| | 3B Ours | 0.8064 | **0.7080** | **0.4713** | **0.7883** | **0.9153** | **0.9323** | **0.9170** | **0.8621** | **0.6812** | **0.7604** |
| | 7B SFT | 0.7253 | 0.6179 | 0.4137 | 0.7374 | 0.9084 | 0.8644 | 0.8749 | 0.7986 | 0.6072 | 0.6892 |
| | 7B Ours | **0.8070** | 0.6997 | 0.4634 | 0.7710 | 0.8926 | 0.9192 | 0.8974 | 0.8341 | 0.6450 | 0.7280 |
| | 3B UB | 0.8469 | 0.7666 | 0.5095 | 0.8300 | 0.9419 | 0.9621 | 0.9484 | 0.9141 | 0.7769 | 0.8383 |
| | 7B UB | 0.8215 | 0.7177 | 0.4736 | 0.7939 | 0.9243 | 0.9378 | 0.9256 | 0.8775 | 0.7076 | 0.7833 |
| Mammary | SAT | 0.1288 | 0.1113 | 0.1929 | 0.4544 | 0.8275 | 0.3711 | 0.5057 | 0.3546 | 0.2034 | 0.2647 |
| | R2Gen | 0.5308 | 0.4489 | 0.3489 | 0.6937 | 0.8826 | 0.7604 | 0.8114 | 0.7103 | 0.4908 | 0.5804 |
| | KMVE | 0.7276 | 0.6414 | 0.4418 | 0.7306 | 0.8420 | 0.8624 | 0.8439 | 0.7563 | 0.5692 | 0.6475 |
| | 3B SFT | 0.7137 | 0.6158 | 0.4173 | 0.7355 | 0.8640 | 0.8518 | 0.8527 | 0.7695 | 0.5708 | 0.6520 |
| | 3B Ours | **0.7654** | 0.6765 | **0.4581** | 0.7581 | 0.8749 | **0.9016** | 0.8828 | 0.8114 | 0.6289 | 0.7064 |
| | 7B SFT | 0.7147 | 0.6110 | 0.4145 | 0.7295 | 0.8471 | 0.8521 | 0.8442 | 0.7585 | 0.5560 | 0.6388 |
| | 7B Ours | 0.7636 | **0.6793** | 0.4580 | **0.7894** | **0.8974** | 0.9012 | **0.8954** | **0.8302** | **0.6584** | **0.7307** |
| | 3B UB | 0.8611 | 0.8071 | 0.5442 | 0.8721 | 0.9500 | 0.9656 | 0.9552 | 0.9238 | 0.8203 | 0.8670 |
| | 7B UB | 0.8804 | 0.8321 | 0.5610 | 0.8930 | 0.9567 | 0.9639 | 0.9580 | 0.9292 | 0.8382 | 0.8791 |
| Liver | SAT | 0.0207 | 0.0191 | 0.1300 | 0.2833 | 0.9964 | 0.6376 | 0.7668 | 0.6340 | 0.3389 | 0.4610 |
| | R2Gen | 0.8518 | 0.7920 | 0.5084 | 0.8519 | 0.9453 | 0.9054 | 0.9206 | 0.8629 | 0.7329 | 0.7927 |
| | KMVE | 0.8803 | 0.8288 | 0.5271 | 0.8660 | 0.8345 | 0.8286 | 0.8104 | 0.7646 | 0.7116 | 0.7291 |
| | 3B SFT | 0.8724 | 0.8054 | 0.5148 | 0.8410 | 0.9192 | 0.9035 | 0.9050 | 0.8386 | 0.7206 | 0.7743 |
| | 3B Ours | 0.9028 | 0.8520 | 0.5538 | 0.8823 | **0.9407** | **0.9440** | **0.9390** | **0.8951** | **0.8022** | **0.8449** |
| | 7B SFT | 0.8537 | 0.7721 | 0.4957 | 0.8069 | 0.9156 | 0.8929 | 0.8976 | 0.8286 | 0.6940 | 0.7544 |
| | 7B Ours | **0.9105** | **0.8613** | **0.5642** | **0.8888** | 0.9394 | 0.9332 | 0.9330 | 0.8838 | 0.7942 | 0.8356 |
| | 3B UB | 0.9326 | 0.9016 | 0.6033 | 0.9271 | 0.9695 | 0.9650 | 0.9654 | 0.9391 | 0.8792 | 0.9070 |
| | 7B UB | 0.9440 | 0.9158 | 0.6195 | 0.9359 | 0.9740 | 0.9684 | 0.9693 | 0.9460 | 0.8942 | 0.9185 |
| Ovary | SAT | 0.0821 | 0.0648 | 0.1579 | 0.3592 | 0.8728 | 0.2852 | 0.4164 | 0.2730 | 0.1487 | 0.1992 |
| | R2Gen | 0.3088 | 0.1872 | 0.2135 | 0.4356 | 0.7198 | 0.6148 | 0.6354 | 0.4908 | 0.2358 | 0.3357 |
| | KMVE | 0.6012 | 0.4235 | 0.3189 | 0.5492 | 0.7117 | 0.6462 | 0.6597 | 0.5163 | 0.2618 | 0.3637 |
| | 3B SFT | 0.5180 | 0.3468 | 0.2813 | 0.4971 | 0.6479 | 0.6134 | 0.6177 | 0.4646 | 0.2243 | 0.3198 |
| | 3B Ours | 0.6310 | 0.4648 | 0.3416 | 0.5803 | 0.7966 | **0.7740** | 0.7735 | 0.6505 | **0.3416** | 0.4673 |
| | 7B SFT | 0.5514 | 0.3714 | 0.2929 | 0.5105 | 0.6493 | 0.6508 | 0.6350 | 0.4839 | 0.2196 | 0.3224 |
| | 7B Ours | **0.6424** | **0.4800** | **0.3497** | **0.5950** | **0.8032** | 0.7580 | **0.7680** | 0.6411 | 0.3301 | 0.4554 |
| | 3B UB | 0.6624 | 0.5167 | 0.3652 | 0.6256 | 0.9352 | 0.8972 | 0.9265 | 0.7695 | 0.4576 | 0.5888 |
| | 7B UB | 0.6666 | 0.5217 | 0.3692 | 0.6389 | 0.9150 | 0.8859 | 0.9117 | 0.7517 | 0.4542 | 0.5799 |

As shown in Table 2, our method enhances clinical accuracy and semantic consistency in medical report generation through Concept Instruction Fine-tuning and a Concept Risk Score–based intervention. It improves the key metric HSM by 9.13% and achieves state-of-the-art performance across all four organs on both the 3B and 7B models by correcting just one Anatomical-Finding Concept Unit with high error risk per report. Notably, on certain smaller organ-specific datasets, the 3B model slightly outperforms the 7B variant, possibly because the larger model's higher capacity leads to overfitting when training data is limited. This minimal intervention is clinically feasible and enables efficient, trustworthy physician–AI collaboration.

## 4.3 CONCEPT RISK SCORE PRIORITIZES CLINICALLY HIGH-LEVERAGE INTERVENTION

We evaluate CRS on 100 randomly sampled thyroid ultrasound cases from the validation set. Correcting only the top-3 CRS-ranked anatomical concepts per report yields a statistically significant correlation with performance improvement ($r = 0.2531$, $p < 0.05$) in Figure 4(b). This confirms CRS successfully prioritizes high-leverage intervention points — reducing physician workload while maximizing model behavior improvement. Note that CRS does not rely on model confidence

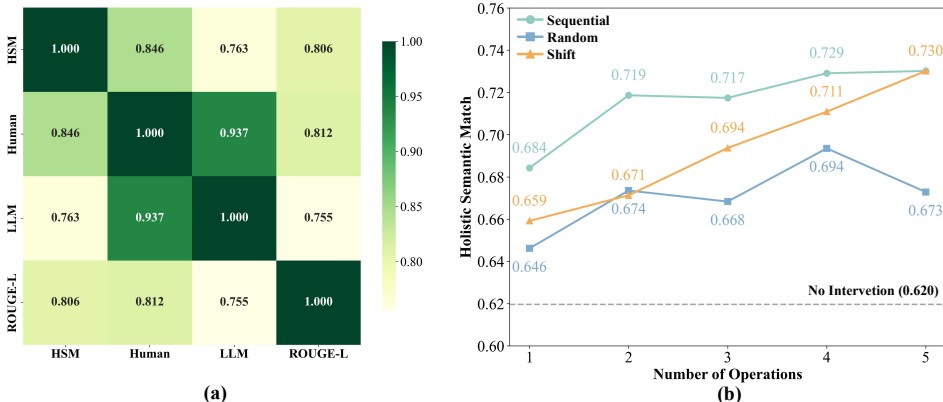

Figure 5: (a) Correlations between automatically computed metrics and human subjective judgments as well as LLM evaluations. (b) Evaluate the impact of varying numbers and orders of finding concepts on intervention effectiveness.

or output probability. It measures behavioral inconsistency under perturbation — a more clinically meaningful signal for feedback prioritization.

## 4.4 ABLATION STUDY OF DIFFERENT UNCERTAINTY SELECTION STRATEGIES

As shown in Table 3, under the Concept Instruction Tuning baseline, we evaluate various uncertainty-based intervention strategies at the anatomical concept level by computing uncertainty over the descriptive phrase associated with each concept. Methods such as Semantic Entropy (Kuhn et al., 2023) and VL-Uncertainty (Zhang et al., 2024) (details in § 2.3) yield varying degrees of improvement. Our proposed Concept Risk Score–based intervention (+ours) achieves the best performance across all metrics, with HSM reaching 0.7604, and significantly outperforms all alternatives. On core clinical metric HSM, our method outperforms the second-best approach by 4.93%—indicating that the Concept Risk Score better

Table 3: Performance of interventions selected by different uncertainty estimation methods under the Concepts Instruction Tuning condition.

| Type | BLEU-4 | ROUGE-L | HSM |
|---|---|---|---|
| Concepts Instruction Tuning | 0.6230 | 0.6985 | 0.6282 |
| +Random | 0.6532 | 0.7365 | 0.6907 |
| +Semantic Entropy | 0.6576 | 0.7436 | 0.7086 |
| +VL-uncertainty | 0.6672 | 0.7488 | 0.7111 |
| +ours | **0.7080** | **0.7883** | **0.7604** |
| Upper Bound | 0.7666 | 0.8300 | 0.8383 |

identifies concepts with high error risk for targeted intervention, yielding reports with improved semantic fidelity. Its performance also closely approaches the theoretical upper bound, confirming the effectiveness of our intervention mechanism. All baseline uncertainty methods are adapted to operate on anatomical concept–level text spans for fair comparison with CRS.

## 4.5 HUMAN ANALYSIS OF CONCEPTS METRIC

We conducted human and LLM evaluations to assess whether our semantic similarity metric, HSM, aligns with human and model judgments. For 100 randomly sampled medical reports, we presented ground-truth and model-generated texts to radiologists and a large language model (DeepSeek-V3.1) (DeepSeek-AI, 2024) to rate semantic similarity (templates and criteria in Appendix § A.5). As shown in Figure 5(a), HSM achieves Spearman correlations of 0.846 with human judgments and 0.763 with LLM ratings, both statistically significant($p < 0.05$), validating HSM as a reliable proxy for clinically meaningful semantic similarity.

## 4.6 ROBUSTNESS TO INCOMPLETE OR DISORDERED FINDING INPUTS

To mimic how clinicians may provide incomplete or ambiguously ordered findings during interactive reporting, we evaluate robustness using Anatomy-Finding Concept Units, where each anatomical concept is paired with exactly five finding concepts. The intervention input is formed by selecting and/or reordering a subset of these findings.

We test three perturbations on 128 validation samples with complete AFCUs: (1) **Sequential-$m$**: first $m$ finding concepts in original order; (2) **Random-$m$**: $m$ randomly selected finding concepts in random order; (3) **Shift-$n$**: full sequence cyclically shifted left by $n$ positions (Shift-5 recovers the original order as a control). As shown in Figure 5(b), performance improves with more findings, yet even minimal inputs (e.g., $m = 1$) outperform no intervention. Sequential inputs consistently surpass Random ones at the same $m$, highlighting the importance of clinical ordering. Shift experiments confirm that performance drops under order perturbations but recovers at Shift-5. These results show our method remains effective under realistic clinician uncertainty—supporting its practical use in interactive report generation.

### 4.7 IMPACT OF MULTIPLE CRS CONCEPT INJECTION

To explore the effect of simultaneously correcting multiple uncertain concepts on report generation, we injected $n$ CRS-ranked concepts into the VLM prompt for 50 liver report cases. As shown in Figure 6, CRS-guided concept injection rapidly improves report quality—as measured by BLEU-1/4 and ROUGE-L—validating CRS's ability to prioritize high-error-risk, clinically informative concepts. However, performance gains diminish with further concept additions. This saturation likely stems from two factors: (1) the remaining concepts carry lower clinical relevance or error risk, and (2) the VLM is not specifically trained to effectively utilize prompts augmented with many additional terms. These findings suggest that selectively integrating the most critical concepts identified by CRS is more effective than exhaustively incorporating numerous concepts. Moreover, our CRS measures error risk—low-risk concepts are often correct (differing only in phrasing)—so omitting them is clinically acceptable. Error accumulation concerns can also be alleviated by correcting multiple high-risk AFCUs.

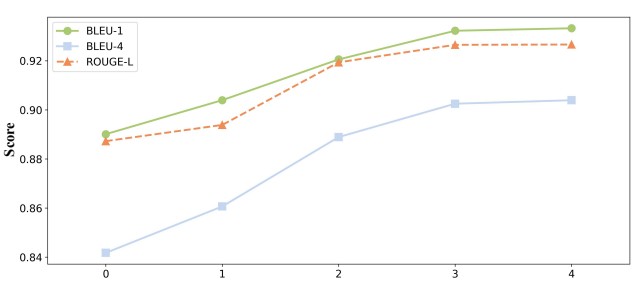

Figure 6: Performance of report generation as more CRS-ranked concepts are injected.

## 5 CONCLUSION

We shift medical report generation from pursuing autonomous accuracy toward effective human–AI collaboration by introducing the RCL-3, a bidirectional paradigm in which VLMs proactively flag content with high error risk and physicians intervene only on the most critical elements. Central to this approach is the Anatomy-Finding Concept Unit, a clinically grounded atomic unit that enables precise, efficient interaction. Leveraging information bottleneck–based concept compression, concept-level instruction tuning for feedback integration, and the Concept Risk Score to prioritize high-impact corrections, our method achieves a 9.13% average gain in Holistic Semantic Match, a clinically aligned metric strongly correlated with human judgment. This demonstrates that minimal, targeted physician input can substantially improve report quality, paving the way for trustworthy, efficient AI-assisted clinical reporting.

### ETHICS STATEMENT

We use publicly available medical report datasets for thyroid, mammary, and liver. The private ovary dataset has received ethical approval; details are in the Appendix. Anonymized details (e.g., ethics approval numbers and data collection sites) will be released upon acceptance.

## REPRODUCIBILITY STATEMENT

The complete code will be released upon paper acceptance. Additionally, all key details necessary for reproducibility—including hyperparameters, training procedures, concept construction methods and examples, LLM prompting templates, and samples from the training and test datasets—are thoroughly described in the main text and the Appendix.

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

# A APPENDIX

- A.1: LLM Usage
- A.2: Detailed Content of Private Data
- A.3: Construction of Anatomical and Finding Concepts
- A.4: Examples of Concept Instruction Tuning
- A.5: Concept-based Metrics Computation
- A.6: Experiment Details

- A.7: Visual Occlusion using Grid-Based Random Mask

- A.8: Case Analysis of Similarity Metrics Across Sentence Embedding Models

- A.9: Comparison of Report Examples Generated by Different Methods

- A.10: Comparison with other Medical VLMs

- A.11: Additional Data

- A.12: Limitations and Societal Impact

- A.13: Stability of HSM Correlation with Traditional Metrics Across Anatomical Organs

- A.14: Ablation Study of Concept Instruction Tuning and CRS

- A.15: Enhancing Clinical Efficacy on the English IU-Xray Dataset

- A.16: Human-in-the-Loop Evaluation

- A.17: Language Presentation and Accessibility

- A.18: Theoretical Analysis of the Concept Risk Score

- A.19: AFCU Enables Clinically Grounded and Fine-Grained Human–AI Interaction

- A.20: HSM Aligns Evaluation with Clinical Reasoning

- A.21: Ablation on Correction Methods Using Zero-shot LLM

## A.1 LLM USAGE

During the preparation of this manuscript, large language models (LLMs) were employed in a limited and auxiliary capacity. Specifically, their usage was restricted to the following three aspects: (1) checking grammar and expression at the sentence level, thereby providing local linguistic refinement; (2) performing global polishing after the draft was completed, ensuring that the overall exposition conforms to idiomatic English usage.

At no stage were LLMs used for generating research ideas, developing arguments, or modifying the substantive content of this work. Their sole role was to assist in enhancing the clarity and effectiveness of communication.

## A.2 DETAILED CONTENT OF PRIVATE DATA

As shown in Table 4 and 5, this study constructs an ovarian lesion ultrasound imaging dataset based on real clinical medical reports from a single center. The dataset comprises 831 pathologically confirmed cases, covering non-neoplastic pelvic masses, benign tumors, borderline tumors, advanced-stage ovarian cancer, early-stage ovarian cancer, and confounding cancer types, corresponding to a total of 1,570 two-dimensional grayscale ultrasound images. Table 5 further demonstrates that the number of images per case follows a natural distribution, authentically reflecting the individualized nature of image acquisition in clinical ultrasound examinations — in contrast to previous studies that often relied on fixed numbers of one or two images per case.

Data collection and usage have been approved by the institutional ethics review board, in compliance with medical research standards. To support transparent peer review, the ethics approval number will be disclosed during the non-anonymous review phase.

Table 4: Case and Image Category Distribution in the Ovarian Ultrasound Dataset

| Statistics | Cases | Case Ratio | Images | Image Ratio |
|---|---|---|---|---|
| Non-neoplastic Pelvic Mass | 289 | 34.8% | 514 | 32.7% |
| Confounding Cancer Types | 61 | 7.3% | 117 | 7.5% |
| Borderline Tumors | 119 | 14.3% | 250 | 15.9% |
| Benign Tumors | 186 | 22.4% | 328 | 20.9% |
| Advanced-stage Ovarian Cancer | 97 | 11.7% | 203 | 12.9% |
| Early-stage Ovarian Cancer | 79 | 0.095 | 158 | 10.1% |
| Total | 831 | - | 1570 | - |

Table 5: Distribution of Image Counts per Case in the Ovarian Ultrasound Dataset

| Image Number | Cases | Case Ratio |
| --- | --- | --- |
| 1 | 243 | 29.2% |
| 2 | 460 | 55.4% |
| 3 | 107 | 12.9% |
| 4 | 19 | 2.3% |
| 5 | 2 | 0.2% |
| Total | 831 | - |

A.3 CONSTRUCTION OF ANATOMICAL AND FINDING CONCEPTS

To promote reproducibility and clinical interpretability, we provide essential descriptions of the canonical anatomical concepts (as shown in Table 6), derived through our information bottleneck-based compression pipeline. These include: Template Design and Initial Concept Extraction, Anatomical Concept Clustering via Semantic Similarity, Binding Findings to Canonical Anatomy. In our framework, a "concept" refers precisely to this structured pairing: one anatomical entity with its clinically relevant attributes, enabling precise, targeted physician–AI collaboration without fragmentation or ambiguity.

Table 6: Examples of Anatomy-Finding Concept Units (AFCUs) — minimal, clinically grounded semantic units pairing a canonical anatomical region with its associated descriptive findings.

| Anatomical Concept | Finding Concept |
| --- | --- |
| Thyroid gland | Size and shape are normal
Uniform echo
No clear space-occupying lesion was found
No abnormal blood flow signals were found |
| Bilateral neck | No obvious enlarged lymph nodes |
| Left lobe gland | Multiple nodules |

**Template Design and Initial Concept Extraction**   As shown in Figures 7 and 8, the Chinese and English versions of the template, respectively, are used to guide the LLM in performing initial extraction of Anatomical and Finding Concepts from each report. Each template provides three examples, though only two are displayed in the figures. These examples are static and generalizable to any new findings or datasets, designed to inform the model of the relative relationship between Anatomical Concepts and Finding Concepts. Manual inspection of the experimental results confirmed that the extracted outputs are clinically acceptable.

**Anatomical Concept Clustering via Semantic Similarity**   Medical reports often exhibit diverse phrasings for the same anatomical structure—e.g., "bilateral thyroid lobes," "left and right lobe," and "bilateral glandular tissue" all describe the same region, reflecting stylistic differences rather than clinical distinctions. As illustrated in Table 7, such lexical redundancy is common in free-text reporting. Normalizing these variants into a single canonical term reduces ambiguity, ensuring that AI–physician collaboration focuses on clinical content rather than wording preferences.

Table 7: Examples of anatomical concept normalization: diverse expressions extracted from free-text reports are mapped to a single canonical term through semantic clustering.

| Report No. | Exacted Anatomical Concept | Anatomical Concept after Mapping |
| --- | --- | --- |
| 1 | bilateral thyroid lobes | both lobes of the thyroid gland |
| 2 | left and right lobe | both lobes of the thyroid gland |
| 3 | bilateral glandular tissue | both lobes of the thyroid gland |

**Binding Findings to Canonical Anatomy**   As shown in Table 8, when multiple extracted anatomical concepts (e.g., "thyroid", "glands") refer to the same clinical entity, they are first mapped to a single canonical term (e.g., "thyroid gland"). Crucially, their associated finding concepts — such as

```
提取以下文本中的解剖结构（anatomical concept）和对应的发现（finding concept），并以 JSON 格式输出。每个解剖结构和其发现应以"entity"和"attributions"键值对的形式显
示。确保输出格式正确，每个解剖结构和其发现按顺序列出。

文本输入：
{{输入的文本内容}}

JSON输出：
{{
  "entities": [
    {{
      "entity": "实体名称",
      "attributions": ["属性1", "属性2", ...]
    }},
    ...
  ]
}}

示例1-文本输入：
{{甲状腺:左叶大小为_3DS_，峡部_SCM_，右叶大小为_3DS_，腺体回声欠均匀，左叶下极见一混合回声结节，大小为_2DS_，边界清晰，形态尚规整，内以无回声为主，并可
见低回声区，范围为_2DS_，CDFI示结节周边可记录到星点状血流信号；右叶内见两个低回声结节，大小分别约_2DS_、_2DS_，边界清晰，形态规整。甲状腺周围未见肿大淋
巴结。}}
示例2-json输出：
{{
  "entities": [
    {{
      "entity": "甲状腺左叶",
      "attributions": ["大小为_3DS_"]
    }},
    {{
      "entity": "峡部",
      "attributions": ["_SCM_"]
    }},
    {{
      "entity": "甲状腺右叶",
      "attributions": ["大小为_3DS_", "腺体回声欠均匀"]
    }},
    {{
      "entity": "左叶结节",
      "attributions": ["下极", "混合回声", "大小为_2DS_", "边界清晰", "形态尚规整", "内以无回声为主", "可见低回声区", "范围为_2DS_", "结节周边可记录到星点状血流信号"]
    }},
    {{
      "entity": "右叶结节",
      "attributions": ["低回声", "大小分别约_2DS_、_2DS_", "边界清晰", "形态规整"]
    }},
    {{
      "entity": "甲状腺周围",
      "attributions": ["未见肿大淋巴结"]
    }}
  ]
}}

示例2-文本输入：
{{甲状腺大小形态如常，腺体回声均匀，未见明确占位性病变，CDFI示腺体内未见异常血流信号。双侧颈部未见明显肿大淋巴结。}}
示例3-json输出：
{{
  "entities": [
    {{
      "entity": "甲状腺",
      "attributions": ["大小形态如常"]
    }},
    {{
      "entity": "腺体",
      "attributions": ["回声均匀", "未见明确占位性病变", "未见异常血流信号"]
    }},
    {{
      "entity": "双侧颈部",
      "attributions": ["未见明显肿大淋巴结"]
    }}
  ]
}}

现在输入要分析的文本，请给出符合要求的json，不要附带别的信息。输入文本：{{{finding_text}}}
```

Figure 7: Chinese prompt template with in-context examples guiding LLMs to extract anatomical and finding concepts in a clinically structured format.

```
Extract the anatomical concepts and corresponding findings from the following text and output them in JSON format. Each anatomical concept and its corresponding finding should be
presented as a key-value pair of "entity" and "attributions". Ensure that the output is well-formatted and that each anatomical concept and its corresponding finding are listed in order.
Text Input: {{Input text content}}

JSON Output:
{{
"entities": [
{
"entity": "Entity Name",
"attributions": ["Attribute 1", "Attribute 2", ...]
},
...
]
}}

Example 1 - Text Input:
{{Thyroid: Left lobe size is 3DS, isthmus SCM, right lobe size is 3DS, gland echoes are uneven, left lobe lower pole shows a mixed echo nodule, size is 2DS, boundary clear, shape still regular,
mainly anechoic inside, low echo areas visible, range is 2DS, CDFI shows spot-like blood flow signals around the nodule; two low-echo nodules in the right lobe, sizes about 2DS and 2DS,
boundary clear, shape regular. No enlarged lymph nodes around the thyroid.}}

Example 2 - JSON Output:
{{
"entities": [
{
"entity": "Left thyroid lobe",
"attributions": ["Size is 3DS"]
},
{
"entity": "Isthmus",
"attributions": ["SCM"]
},
{
"entity": "Right thyroid lobe",
"attributions": ["Size is 3DS", "Gland echoes are uneven"]
},
{
"entity": "Left lobe nodule",
"attributions": ["Lower pole", "Mixed echo", "Size is 2DS", "Boundary clear", "Shape still regular", "Mainly anechoic inside", "Low echo areas visible", "Range is 2DS", "Spot-like blood flow
signals recorded around the nodule"]
},
{
"entity": "Right lobe nodule",
"attributions": ["Low echo", "Sizes about 2DS and 2DS", "Boundary clear", "Shape regular"]
},
{
"entity": "Around the thyroid",
"attributions": ["No enlarged lymph nodes"]
}
]
}}

Example 2 - Text Input:
{{Thyroid size and shape are normal, gland echoes are uniform, no definite space-occupying lesions, CDFI shows no abnormal blood flow signals in the gland. No obvious enlarged lymph
nodes in both sides of the neck.}}

Example 3 - JSON Output:
{{
"entities": [
{
"entity": "Thyroid",
"attributions": ["Size and shape are normal"]
},
{
"entity": "Gland",
"attributions": ["Echoes uniform", "No definite space-occupying lesions", "No abnormal blood flow signals"]
},
{
"entity": "Both sides of the neck",
"attributions": ["No obvious enlarged lymph nodes"]
}
]
}}
Now input the text to be analyzed, and provide the corresponding JSON. Do not include any additional information. Input text: {{finding_text}}
```

Figure 8: The English version of Chinese prompt template with in-context examples guiding LLMs to extract anatomical and finding concepts in a clinically structured format.

"size normal" or "no abnormal blood flow" — must then be aggregated under this unified anatomical anchor. This process, termed Binding Findings to Canonical Anatomy, ensures that no clinical information is lost during compression and that each Anatomy-Finding Concept Unit (AFCU) remains semantically complete and clinically grounded — forming the atomic unit for physician-AI interaction.

Table 8: Example of binding finding concepts to canonical anatomical concepts after semantic mapping — consolidating findings under unified anatomy terms for structured AFCU representation.

| | Anatomical Concept | Finding Concept |
|---|---|---|
| | thyroid | Size and shape are normal |
| Before Mapping | glands | Uniform echo
No clear space-occupying lesion was found
No abnormal blood flow signals were found |
| After Mapping and Binding | thyroid gland | Size and shape are normal
Uniform echo
No clear space-occupying lesion was found
No abnormal blood flow signals were found |

## A.4 EXAMPLES OF CONCEPT INSTRUCTION TUNING

The Anatomy-Finding Concept Unit (AFCU), which pairs an anatomical concept with one or more finding concepts, serves as additional contextual information within the instruction for generating medical reports. As illustrated in Figure 9, the first example represents the traditional fine-tuning method without AFCUs, whereas the second and third examples showcase our proposed Concept Instruction Tuning. Notably, a single anatomical concept may correspond to multiple finding concepts; during both fine-tuning and testing, all relevant findings are included to comprehensively explore our framework's capabilities. Furthermore, Figure 5(b) delves into how varying the number and order of finding concepts impacts model performance, revealing consistent improvements across different configurations. This approach not only enhances the accuracy of generated reports but also ensures that no clinically significant detail is overlooked.

## A.5 CONCEPT-BASED METRICS COMPUTATION

As shown in Figure 11, we only use the LLM during the dictionary initialization phase. In every subsequent inference, particularly when evaluating on a new test dataset, finding concepts for an anatomical concept can be approximated simply by extracting the text segment between two consecutive anatomical concepts, following the fixed subject–verb–object syntactic structure commonly used in Chinese. Then, by comparing the sets of anatomical concepts, we identify the presence of entities and compute the similarity of finding concepts only for corresponding anatomical concepts to evaluate fine-grained semantic similarity. As illustrated in the example, our metrics can still make more nuanced and accurate judgments even when conventional NLG metrics (e.g., ROUGE-L) or sentence-level embedding cosine similarity scores (STS) are high. Moreover, as shown in Figure 5(b), we demonstrate that our HSM metric exhibits a strong correlation with both LLM-based and human evaluations (the LLM evaluation prompt template is provided in Figure 10, and human annotators used the same rating criteria). Discussion on extracting finding concepts for English datasets can be found in Appendix § A.12.

## A.6 EXPERIMENT DETAILS

As shown in Table 9, we compare the fine-tuning setups for 3B and 7B VLMs, with and without Concept Instruction Tuning. In all cases, the visual encoder and projector are fully fine-tuned, while the LLM is updated only via LoRA. For concept instruction tuning, we add synthetic AFCU examples to the original data. On the Thyroid dataset, this expands the training set from 1,719 samples to 8,884 by including 7,165 generated examples. Training with the AFCU-augmented 3B model on a single A100 40GB GPU takes 3 hours and 31 minutes; for the 7B model, it takes 4 hours and 51 minutes. All model outputs are generated using beam search with a beam width of 3. All hyperparameters — including batch size, LoRA rank, learning rate, and training epochs — are kept consistent to ensure fair comparison and reproducibility. For other baseline methods, we train for 100 epochs to ensure convergence and report the best-performing checkpoint.

```
{
"messages": [
    {
        "role": "user",
        "content": "<image><image>By analyzing the images and using the relevant information to generate a detailed medical report. "
    },
    {
        "role": "assistant",
        "content": "After partial thyroidectomy, the morphology was abnormal, and multiple cystic solid mixed echo nodules could be seen in the left lobe. The largest
was located in the middle section, with a size of about 2DS, with clear boundaries and regular morphology. CDFI shows that the star-like blood flow signal can be
detected around the periphery. The echo of the rest of the parenchyma is still uniform, and CDFI shows that the blood flow signal is roughly normal. No enlarged lymph
nodes were found around the thyroid gland and beside the large blood vessels on both sides of the neck."
    }
    ],
"image_path": [
        "183961_1.jpeg",
        "183961_2.jpeg"
    ]
},

{
"messages": [
    {
        "role": "user",
        "content": "<image><image>By analyzing the images and using the relevant information to generate a detailed medical report. Information: Present thyroid,
after partial resection, illegal shape, no enlarged lymph nodes were found."
    },
    {
        "role": "assistant",
        "content": "After partial thyroidectomy, the morphology was abnormal, and multiple cystic solid mixed echo nodules could be seen in the left lobe. The largest
was located in the middle section, with a size of about 2DS, with clear boundaries and regular morphology. CDFI shows that the star-like blood flow signal can be
detected around the periphery. The echo of the rest of the parenchyma is still uniform, and CDFI shows that the blood flow signal is roughly normal. No enlarged lymph
nodes were found around the thyroid gland and beside the large blood vessels on both sides of the neck."
    }
    ],
"image_path": [
        "183961_1.jpeg",
        "183961_2.jpeg"
    ]
},

{
"messages": [
    {
        "role": "user",
        "content": "<image><image>By analyzing the images and using the relevant information to generate a detailed medical report. Information: Present bilateral
neck, no enlarged lymph nodes were found."
    },
    {
        "role": "assistant",
        "content": "After partial thyroidectomy, the morphology was abnormal, and multiple cystic solid mixed echo nodules could be seen in the left lobe. The largest
was located in the middle section, with a size of about 2DS, with clear boundaries and regular morphology. CDFI shows that the star-like blood flow signal can be
detected around the periphery. The echo of the rest of the parenchyma is still uniform, and CDFI shows that the blood flow signal is roughly normal. No enlarged lymph
nodes were found around the thyroid gland and beside the large blood vessels on both sides of the neck."
    }
    ],
"image_path": [
        "183961_1.jpeg",
        "183961_2.jpeg"
    ],
}
```

Figure 9: Illustration of Concept Instruction Tuning, showing baseline fine-tuning versus our method with multiple finding concepts paired to a single anatomical concept.

Table 9: Fine-tuning Hyperparameters for Qwen2.5-VL: Standard SFT vs AFCU-Augmented Instruction Tuning. V denotes Vision Encoder, P denotes Projector.

|  | SFT-3B | SFT-7B | AFCU-SFT-3B | AFCU-SFT-7B |
|---|---|---|---|---|
| Trainable module | V+P+LLM(LoRA) | V+P+LLM(LoRA) | V+P+LLM(LoRA) | V+P+LLM(LoRA) |
| Training data | origin | origin | origin+AFCUI | origin+AFCUI |
| Learning rate | 1e-4 | 5e-5 | 1e-4 | 5e-5 |
| batch_size | 4 | 2 | 4 | 2 |
| grad_accum_steps | 16 | 32 | 16 | 32 |
| Effective batch size | 64 | 64 | 64 | 64 |
| Warmup ratio | 0.20 | 0.20 | 0.20 | 0.20 |
| Training epochs | 20 | 20 | 5 | 5 |
| LoRA rank | 32 | 32 | 32 | 32 |
| LoRA alpha | 64 | 64 | 64 | 64 |
| LoRA dropout | 0.1 | 0.2 | 0.1 | 0.2 |
| Weight decay | 0.05 | 0.05 | 0.05 | 0.05 |
| Warmup ratio | 0.20 | 0.2 | 0.20 | 0.2 |
| Max gradient norm | 2.0 | 1.0 | 2.0 | 1.0 |
| Trainable Params | 5.12 GB | 5.21 GB | 5.12 GB | 5.21 GB |
| All Params | 16.61 GB | 33.55 GB | 16.61 GB | 33.55 GB |

请阅读以下两个医学报告并评估它们之间的语义相似度。根据报告中的疾病诊断、症状描述、检查结果等内容，给出一个从0到10的相似度评分，并按以下标准给出评分：
  0分：完全不相似，报告内容没有任何相似之处。
  1分：极低相似度，报告内容几乎完全不同，只有个别术语或非常小的元素相似。
  2分：非常低相似度，报告有少量共同的术语或描述，但在诊断、治疗等方面差异较大。
  3分：低相似度，报告在某些术语或症状上有相似之处，但差异明显，整体内容不一致。
  4分：较低相似度，报告在某些部分存在共同点，但差异较多，整体结构和重点不同。
  5分：中等相似度，报告在某些方面相似（如症状或治疗方法），但在一些重要领域（如诊断）存在差异。
  6分：高相似度，报告在多个方面（如疾病描述、治疗方案等）相似，但细节或某些领域存在差异。
  7分：非常高相似度，报告在大部分内容上相似，仅在少数细节上存在差异。
  8分：极高相似度，报告在核心内容上高度一致，仅在措辞或非常细微的地方存在差异。
  9分：几乎相同，报告内容几乎完全一致，差异极其微小，仅在一些不重要的细节上有所不同。
  10分：强烈接收，报告完全相同，内容完全一致，无法区分。
  请以'得分；理由'的形式提供你的评分和解释，指出报告内容的相似点和差异。
  报告1：
  {report1}
  报告2：
  {report2}

Please read the following two medical reports and assess their semantic similarity. Based on the reports' content, such as disease diagnoses, symptom descriptions, and test results, assign a similarity score from 0 to 10, using the following criteria:
0: Totally dissimilar; the reports share no similarities at all.
1: Extremely low similarity; the reports are almost completely different, with only a few terms or very minor elements being similar.
2: Very low similarity; the reports share a few terms or descriptions, but differ significantly in terms of diagnoses, treatments, and other areas.
3: Low similarity; the reports share some similarities in terms or symptoms, but the differences are significant, and the overall content is inconsistent.
4: Low similarity; the reports share some similarities, but differ significantly, with different overall structure and focus.
5: Moderately similar; the reports are similar in some areas (such as symptoms or treatments), but differ in some important areas (such as diagnoses).
6: Highly similar; the reports are similar in many areas (such as disease descriptions and treatment plans), but differ in details or certain areas. 7: Very similar. The reports are similar in most respects, with only minor differences.
8: Extremely similar. The reports are highly consistent in their core content, with only minor or minor differences in wording.
9: Almost identical. The reports are almost identical in content, with only minor differences in minor details.
10: Strongly accepted. The reports are identical, with content that is indistinguishable.
Please provide your score and explanation in the form of 'Score; Reason', identifying the similarities and differences in the reports.
  Report 1:
  {report1}
  Report 2:
  {report2}

Figure 10: Prompt template used by the LLM to evaluate semantic similarity between generated and reference medical reports.

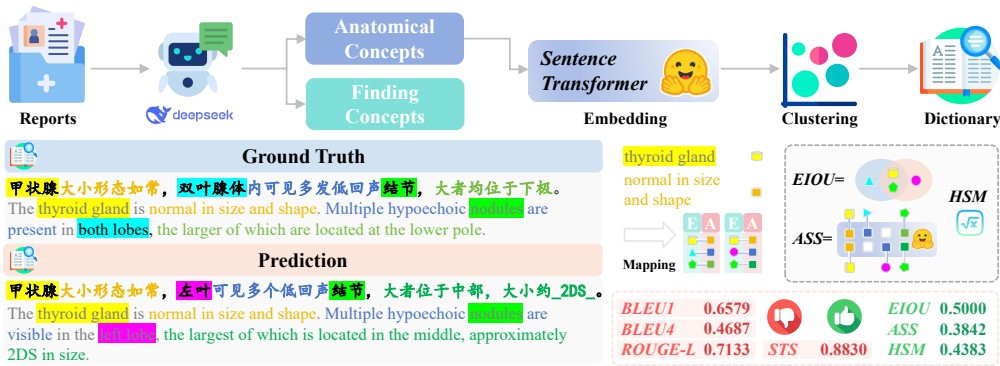

Figure 11: Workflow of anatomical and finding concepts extraction and concept-based semantic similarity metric calculation.

## A.7 Visual Occlusion using Grid-Based Random Mask

As illustrated in Figure 12 and Algorithm 1, we employ a grid-based visual occlusion technique to selectively disrupt the semantic content of images through random masking.

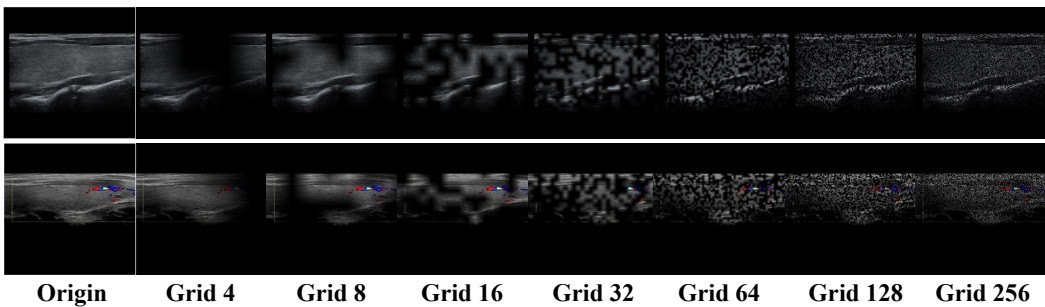

| **Origin** | **Grid 4** | **Grid 8** | **Grid 16** | **Grid 32** | **Grid 64** | **Grid 128** | **Grid 256** |

Figure 12: Visual examples of grid-based random masking with grid sizes ranging from 4 to 256 (7 levels in total).

---

**Algorithm 1** Apply Grid-Based Random Mask to Image

---

**Require:** Image $I \in \mathbb{R}^{H \times W \times 3}$, grid size $G$, mask probability threshold $\tau$
**Ensure:** Masked image $I_{\text{masked}}$
1: $W, H \leftarrow \text{width}(I), \text{height}(I)$
2: $c_w \leftarrow \lceil W/G \rceil, \quad c_h \leftarrow \lceil H/G \rceil$              ▷ Cell size
3: $U_w \leftarrow (G+1) \cdot c_w, \quad U_h \leftarrow (G+1) \cdot c_h$        ▷ Upsampled mask size
4: Initialize random grid mask: $M_{\text{grid}} \in \mathbb{R}^{G \times G}$
5: **for** $i = 0$ to $G - 1$ **do**
6:      **for** $j = 0$ to $G - 1$ **do**
7:          $M_{\text{grid}}[i, j] \leftarrow \begin{cases} 1 & \text{if Uniform}(0, 1) < \tau \\ 0 & \text{otherwise} \end{cases}$
8:      **end for**
9: **end for**
10: $M_{\text{up}} \leftarrow \text{Resize}(M_{\text{grid}}, (U_w, U_h))$           ▷ Bilinear interpolation
11: Sample random offset: $o_w \sim \text{Uniform}(0, c_w - 1), \ o_h \sim \text{Uniform}(0, c_h - 1)$
12: $M \leftarrow M_{\text{up}}[o_h : o_h + H, \ o_w : o_w + W]$        ▷ Crop to original size
13: Normalize image: $\hat{I} \leftarrow I/255.0$
14: Broadcast mask: $M_{\text{rgb}} \leftarrow \text{Stack}(M, M, M)$         ▷ Shape: $H \times W \times 3$
15: $I_{\text{masked}} \leftarrow (\hat{I} \odot M_{\text{rgb}}) \times 255$
16: Convert $I_{\text{masked}}$ to `uint8`
17: **return** $I_{\text{masked}}$

---

## A.8 Case Analysis of Similarity Metrics Across Sentence Embedding Models

We conduct a case study by comparing the generated report shown in Figure 11 with its ground-truth report. Specifically, our proposed FSS metric leverages Sentence Transformer (Reimers & Gurevych, 2019) embeddings to evaluate fine-grained semantic alignment between generated and reference findings. As shown in Table 10, we evaluate multiple multilingual embedding models and observe that FSS and HSM yield consistently lower but more discriminative scores compared to conventional NLG metrics like BLEU-1 and ROUGE-L. Importantly, the relative ranking of models under FSS closely matches that of the baseline model distiluse-base-multilingual-cased-v1 (Reimers & Gurevych, 2019) — which we have shown in the main text to correlate strongly with human judgment — demonstrating that FSS is robust across different embedding backbones. Notably, this cross-model consistency stems from FSS's design: it computes similarity only between finding concepts that are explicitly bound to the same canonical anatomical concept, enabling precise, structure-aware evaluation that generalizes well across languages and embedding architectures.

Table 10: Comparison of multilingual Sentence Transformer models (Reimers & Gurevych, 2020; 2019). STS scores are computed by encoding the full report sentences into embeddings and measuring cosine similarity; FSS and HSM are our fine-grained metrics that operate only on anatomy-aligned finding concepts.

| Model | STS | BLEU-1 | BLEU-4 | ROUGE-L | AIOU | FSS | HSM |
|---|---|---|---|---|---|---|---|
| distiluse-base-multilingual-cased-v1 | 0.8830 | | | | | 0.3842 | 0.4383 |
| distiluse-base-multilingual-cased-v2 | 0.7826 | | | | | 0.3725 | 0.4316 |
| paraphrase-multilingual-mpnet-base-v2 | 0.9180 | 0.6579 | 0.4687 | 0.7133 | 0.5000 | 0.3781 | 0.4348 |
| use-cmlm-multilingual | 0.8206 | | | | | 0.3850 | 0.4387 |
| msmarco-distilbert-multilingual-en-de-v2-tmp-lng-aligned | 0.9112 | | | | | 0.4093 | 0.4524 |
| distiluse-base-multilingual-cased | 0.7826 | | | | | 0.3725 | 0.4316 |

A.9 COMPARISON OF REPORT EXAMPLES GENERATED BY DIFFERENT METHODS

As shown in Figure 13, we compare report generation on mammary and liver examples. Standard SFT often fails to align findings with the correct anatomical concepts, whereas Concept Instruction Tuning (using AFCU) with CRS-based Intervention accurately associates findings with their corresponding anatomical structures, producing more precise and structured reports.

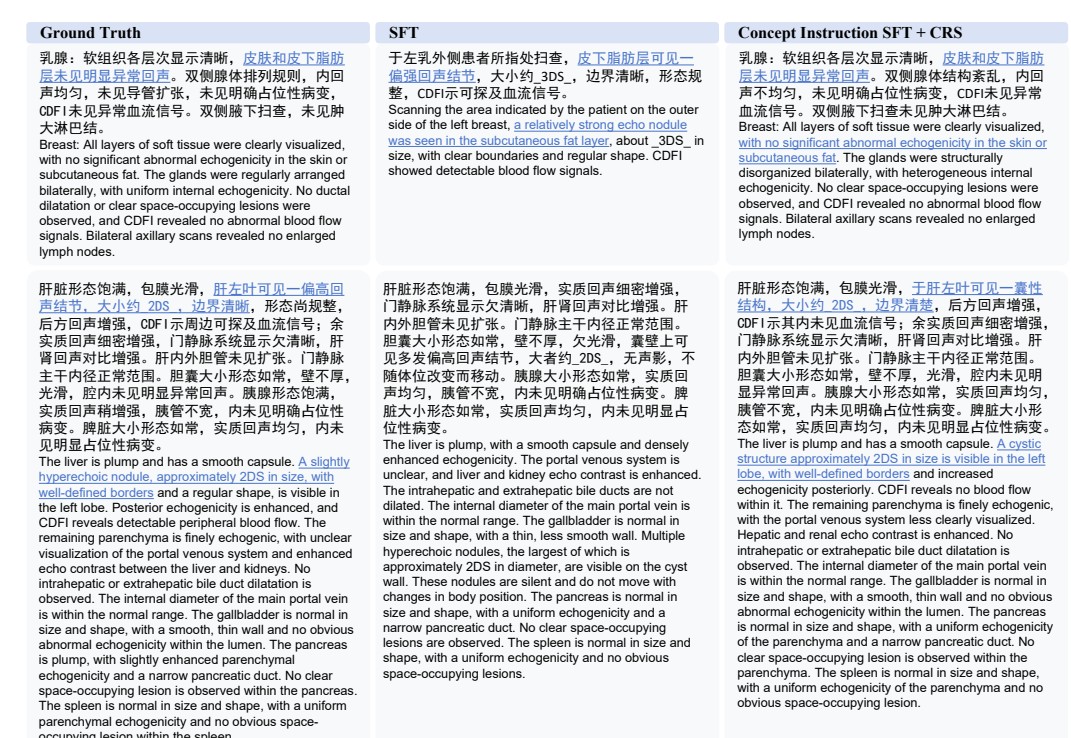

Figure 13: Comparison of report examples generated on the mammary and liver datasets.

A.10 COMPARISON WITH OTHER MEDICAL VLMS

As shown in the Table 11, our bidirectional collaborative framework improves medical VLMs (Xu et al., 2025; Sellergren et al., 2025) on the English thyroid report dataset—translated from a Chinese clinical dataset via Google Translate—with HSM increasing from 0.6445 (SFT) to 0.6777. A key limitation stems from our reliance on machine-translated reports: while our framework operates in English to leverage strong open-source VLMs, subtle clinical semantics in the original Chinese reports—particularly in nuanced finding descriptors (e.g., margin, echogenicity)—may be lost or distorted during translation, weakening the anatomical grounding of AFCUs and limiting the effectiveness of concept-level interventions.

Table 11: Performance comparison of various VLMs on the English thyroid ultrasound report dataset.

| Model | BLEU-1 | BLEU-4 | METEOR | ROUGE-L | Precision | Recall | F1 Score | AIOU | FSS | HSM |
|---|---|---|---|---|---|---|---|---|---|---|
| Lingshu-32B | 0.2207 | 0.0153 | 0.1348 | 0.1573 | 0.5537 | 0.2375 | 0.3123 | 0.2049 | 0.0488 | 0.0968 |
| Qwen2.5-VL-3B | 0.0846 | 0.0055 | 0.0912 | 0.0957 | 0.2484 | 0.1593 | 0.1858 | 0.1187 | 0.0246 | 0.0507 |
| Medgemma-4b | 0.1059 | 0.0075 | 0.1148 | 0.1123 | 0.2412 | 0.1450 | 0.1746 | 0.1125 | 0.0228 | 0.0480 |
| Qwen2.5-VL-3B+SFT | 0.6220 | 0.4367 | 0.3284 | 0.6042 | 0.8541 | 0.8215 | 0.8260 | 0.7281 | 0.5622 | 0.6351 |
| Medgemma-4b+SFT | 0.6248 | 0.4350 | 0.3318 | 0.6039 | 0.8635 | 0.8219 | 0.8311 | 0.7378 | 0.5712 | 0.6445 |
| Medgemma-4b+Ours | **0.6642** | **0.4609** | **0.3558** | **0.6107** | **0.8879** | **0.8676** | **0.8689** | **0.7877** | **0.5916** | **0.6777** |
| Medgemma-4b UB | 0.6642 | 0.4774 | 0.3737 | 0.6534 | 0.9303 | 0.9193 | 0.9194 | 0.8658 | 0.6865 | 0.7668 |

## A.11 ADDITIONAL DATA

Tables 12 and 13 are the full versions of the corresponding tables in the main text (see Tables 1 and 3).

Table 12: Average performance of interventions at different levels after instruction Supervised Fine-Tuning (SFT). Concept-level (AFCU) SFT refers to our proposed Concept Instruction Tuning.

| Type | BLEU-4 | ROUGE-L | F1 Score | AIOU | FSS | HSM |
|---|---|---|---|---|---|---|
| SFT | 0.6341 | 0.7277 | 0.8596 | 0.7752 | 0.5531 | 0.6475 |
| + Phrase-level Intervention | 0.5662 | 0.6775 | 0.8356 | 0.7404 | 0.4962 | 0.5984 |
| + Sentence-level Intervention | 0.6481 | 0.7380 | 0.8191 | 0.7188 | 0.5603 | 0.6280 |
| + Report-level Intervention | **0.8669** | **0.8999** | **0.9455** | **0.9141** | **0.8337** | **0.8681** |
| Concept-level (GPT) SFT | 0.5653 | 0.6434 | 0.8044 | 0.6925 | 0.4606 | 0.5602 |
| + Concept-level (GPT) | 0.5131 | 0.6370 | 0.8145 | 0.7078 | 0.4315 | 0.5470 |
| Concept-level (AFCU) SFT | 0.6230 | 0.6985 | 0.8459 | 0.7543 | 0.5321 | 0.6282 |
| + Concept-level (AFCU) | 0.6604 | 0.7413 | 0.8861 | 0.8157 | 0.6131 | 0.7015 |

Table 13: Performance of interventions selected by different uncertainty estimation methods under the Concepts Instruction Tuning condition.

| Type | BLEU-4 | ROUGE-L | F1 Score | AIOU | FSS | HSM |
|---|---|---|---|---|---|---|
| Concepts Instruction Tuning | 0.6230 | 0.6985 | 0.8459 | 0.7543 | 0.5321 | 0.6282 |
| +Random | 0.6532 | 0.7365 | 0.8797 | 0.8062 | 0.6019 | 0.6907 |
| +Semantic Entropy | 0.6576 | 0.7436 | 0.8909 | 0.8236 | 0.6200 | 0.7086 |
| +VL-uncertainty | 0.6672 | 0.7488 | 0.8903 | 0.8222 | 0.6252 | 0.7111 |
| +ours | **0.7080** | **0.7883** | **0.9170** | **0.8621** | **0.6812** | **0.7604** |
| Upper Bound | 0.7666 | 0.8300 | 0.9484 | 0.9141 | 0.7769 | 0.8383 |

## A.12 LIMITATIONS AND SOCIETAL IMPACT

Although our Concept Risk Score effectively identifies high-risk present-but-misgrounded anatomical concepts, it assumes the VLM can already detect relevant anatomy, making it unable to flag anatomical concepts omitted by the model. Recent works (e.g., knowledge-graph or tree-reasoning models) also use structured concept representations, but primarily to improve internal generation mechanisms. In contrast, our focus is on human–AI collaboration: AFCU and CRS are not part of the generator, but interpretable, actionable units for physician feedback. Future work could integrate such models as backbones within the RCL-3 framework to further enhance collaborative efficiency. Regarding the Holistic Semantic Match (HSM) metric, its Finding Semantic Similarity (FSS) component, which compares finding concepts tied to each anatomical entity, is designed to be language-agnostic. However, implementation is simpler for Chinese reports, which follow a consistent subject–predicate–object structure: the text between consecutive anatomical concepts often directly encodes the associated finding concepts. In contrast, English reports use more varied syntax (e.g., passive voice, embedded clauses), requiring robust Natural Language Processing (NLP) tools for accurate finding extraction. Our current RCL-3 implementation targets only the single anatomical concept with high error risk per report. While this minimizes physician effort and enables rapid correction, it leaves other errors unaddressed. This is a deliberate trade-off that favors high-leverage interventions over exhaustive review.

Table 14: Correlation between HSM and BLEU metrics across different organs, showing both mean scores and Pearson correlation coefficients.

| Organ | Sample Size | HSM Mean | BLEU1 Mean | BLEU4 Mean | HSM–BLEU1 | | HSM–BLEU4 | |
|---|---|---|---|---|---|---|---|---|
| | | | | | R | p-value | R | p-value |
| Liver | 279 | 0.8449 | 0.8854 | 0.8379 | 0.8467 | p<0.001 | 0.8735 | p<0.001 |
| Mammary | 703 | 0.7064 | 0.7228 | 0.6557 | 0.8335 | p<0.001 | 0.8896 | p<0.001 |
| Ovary | 169 | 0.4673 | 0.5901 | 0.4415 | 0.5480 | p<0.001 | 0.6979 | p<0.001 |

Table 15: Ablation study of our method with intervention prompts. "Base + Prompt": uses the original QwenVL-3B model with intervention prompts applied at inference time. "Concept Instruction Tuning" and "CRS" are the two core components of our proposed method.

| Methods | BLEU-1 | BLEU-4 | METEOR | ROUGE-L |
|---|---|---|---|---|
| Base + Prompt | 0.1043 | 0.0301 | 0.1327 | 0.3478 |
| Concept Instruction Tuning + Prompt | 0.6951 | 0.5931 | 0.4342 | 0.7078 |
| Concept Instruction Tuning + Prompt+ CRS | 0.8064 | 0.7080 | 0.4713 | 0.7883 |
| Upper Bound | 0.8469 | 0.7666 | 0.5095 | 0.8300 |

Nevertheless, this focused approach marks a foundational step toward scalable, trust-aware human–AI collaboration. By shifting from full manual revision to concept-level guidance, we reduce workload while preserving oversight. The framework is inherently extensible: as VLMs and CRS improve, the same protocol can support multi-concept or iterative refinement, offering a scalable blueprint for real-world clinical deployment. We hope our findings provide meaningful insights and practical guidance for developing truly effective human–AI collaboration paradigms, ultimately supporting the genuine deployability of medical report generation systems.

### A.13 STABILITY OF HSM CORRELATION WITH TRADITIONAL METRICS ACROSS ANATOMICAL ORGANS

As shown in Figure 14, the correlation between HSM and BLEU metrics is generally strong but varies across organs, reflecting differences in reporting complexity and semantic structure. For Liver and Mammary—organs with more standardized descriptions—the correlations are high (R > 0.83 for BLEU-1, R > 0.87 for BLEU-4), suggesting that n-gram overlap often aligns with semantic correctness. However, for Ovary, the correlation drops notably (R = 0.55 for BLEU-1), indicating that BLEU can be misleading when reports contain fluent but clinically inaccurate phrasing. This variability demonstrates that while BLEU may serve as a rough proxy in simpler cases, HSM offers more stable and clinically meaningful evaluation across diverse anatomical contexts.

### A.14 ABLATION STUDY OF CONCEPT INSTRUCTION TUNING AND CRS

Table 15 presents an ablation study to evaluate the contributions of our key components. Using only intervention prompts with the base QwenVL-3B model ("Base + Prompt") yields poor performance, highlighting the necessity of task-specific adaptation. Incorporating Concept Instruction Tuning dramatically improves all metrics, demonstrating that the intervention capability stems not from the base model itself, but from our fine-tuning strategy. Adding CRS (Concept Refinement Strategy) further boosts performance across the board, confirming that uncertainty-aware, targeted intervention enhances generation quality. The final result approaches the theoretical upper bound—achieved by feeding the full ground-truth report as input—validating the efficacy of our human–AI collaboration framework.

### A.15 ENHANCING CLINICAL EFFICACY ON THE ENGLISH IU-XRAY DATASET

To evaluate clinical utility, we report clinical efficacy (CE) metrics—precision, recall, and F1—on the English-language IU-Xray dataset (Demner-Fushman et al., 2015). As shown in Table 16, our method achieves 0.470 precision, 0.468 recall, and 0.468 F1, substantially outperforming R2Gen (Chen et al., 2020), CVT2Dis (Nicolson et al., 2023), M2KT (Nooralahzadeh et al., 2021), PromptMRG (Jin et al., 2024a), and REVTAF (Zhou et al., 2025) (F1: 0.273). CE metrics are evaluated using the same CheXbert-based (Smit et al., 2020) validation code as in Zhou et al. (2025). The significant gain likely stems from our AFCU-based interaction: when a physician corrects an Anatomy-Finding Concept Unit, the refined clinical finding is directly incorporated into the final

Table 16: Clinical efficacy and language quality on IU-Xray. Our method achieves substantially higher CE metrics (precision, recall, F1) while maintaining competitive METEOR scores versus state-of-the-art approaches.

|  | Year | METEOR | Precision | Recall | F1 |
|---|---|---|---|---|---|
| R2Gen | ACL 2020 | 0.128 | 0.151 | 0.145 | 0.145 |
| CVT2Dis | Artif.Intell.Med 2022 | 0.147 | 0.174 | 0.172 | 0.168 |
| M2KT | MIA 2023 | 0.153 | 0.153 | 0.145 | 0.145 |
| PromptMRG | AAAI 2024 | 0.160 | 0.213 | 0.229 | 0.211 |
| REVTAF | ICCV 2025 | 0.176 | 0.286 | 0.282 | 0.273 |
| Ours | - | 0.178 | 0.470 | 0.468 | 0.468 |

report, leading to more accurate CheXbert labeling. This validates a core claim of our work—the necessity of efficient, concept-level physician intervention for clinically meaningful refinement. Notably, this strong performance is achieved despite our model being primarily trained on Chinese ultrasound data, demonstrating robust multilingual generalization.

### A.16 HUMAN-IN-THE-LOOP EVALUATION

To demonstrate the time savings and reduced workload on physicians achieved by our method, we compared the time required by human annotators under different generation settings, as shown in Table 17. We evaluated 15 samples from the Mammary dataset and measured the average time spent under four configurations: (1) fully human-generated, (2) fully AI-generated, (3) AI-generated with manual checking and editing of one AFCU by a physician, and (4) AI-generated with CRS-based checking and subsequent editing of one AFCU by a physician. The results show that setting (4) reduces the average time by 14.34 seconds compared to setting (3), demonstrating that our CRS indeed has the potential to alleviate the burden on physicians.

Table 17: Average Time (seconds) per Sample under Different Generation Settings on 15 Mammary Cases. Our CRS reduces human effort by 14.31 s compared to manual checking.

| Human-Generated | AI-Generated | Human Checking | CRS | Human Editing | Time (s) |
|---|---|---|---|---|---|
| ✓ |  |  |  |  | 205.95 |
|  | ✓ |  |  |  | 15.16 |
|  | ✓ | ✓ |  | ✓ | 46.10 |
|  | ✓ |  | ✓ | ✓ | 31.76 |

### A.17 LANGUAGE PRESENTATION AND ACCESSIBILITY

All experiments in this work are conducted on a Chinese ultrasound report dataset. Following the practice of prior work such as KMVE (Li et al., 2024a), we retain the original Chinese text in figures to ensure academic fidelity and avoid potential semantic distortion from translation. Nevertheless, to improve accessibility for non-Chinese readers, every Chinese segment is paired with an English translation.

To further enhance readability, we provide English-only versions of all key figures (including Figures 14, 15, 16 and 17).

### A.18 THEORETICAL ANALYSIS OF THE CONCEPT RISK SCORE

The Concept Risk Score is a principled metric for identifying anatomical concepts whose generation by a vision-language model lacks reliable visual grounding. Rather than relying on ad hoc heuristics, CRS is derived from two formal desiderata for robust medical report generation under input perturbations: **Input Sensitivity**: A visually grounded VLM must exhibit dependence on image content; specifically, occlusion of the region corresponding to an anatomical concept $e$ should reduce or eliminate its generation. **Semantic Consistency**: Conditional on the presence of $e$, the associated descriptive findings (e.g., "irregular margin", "hypoechoic") should induce stable semantic representations across perturbed views of the same image.

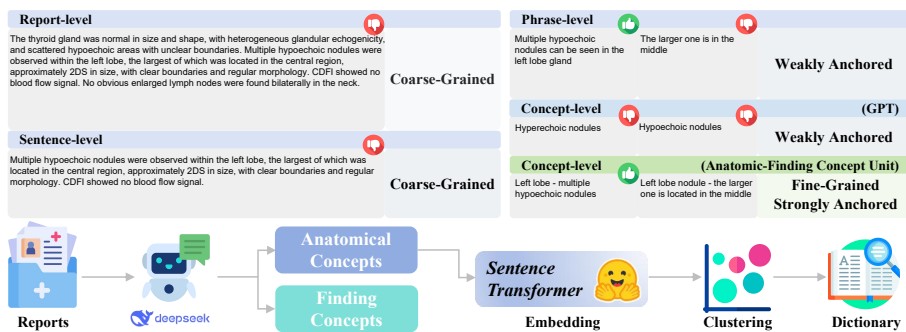

Figure 14: Comparison of Interaction Levels by Semantic Granularity and Anchoring, with Anatomic-Finding Concept Unit Extraction Pipeline (English-only version).

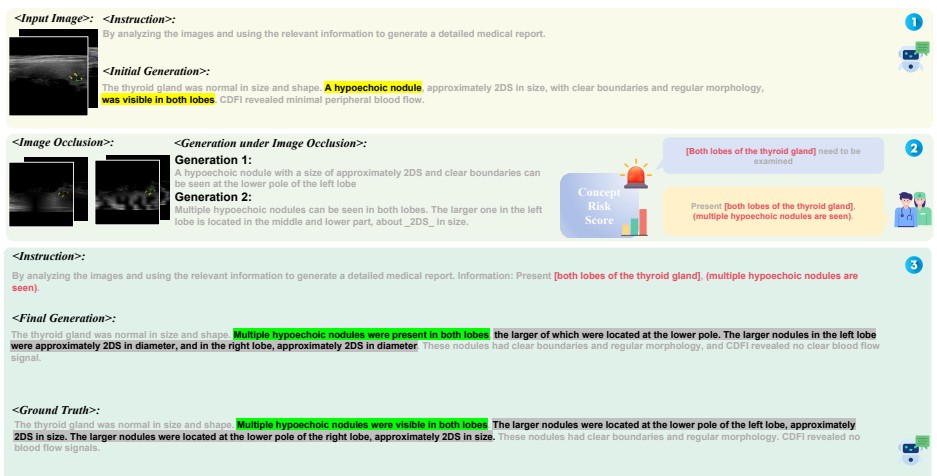

Figure 15: The three main stages of bidirectional human-AI collaborative report generation. Among them, Concept Instruction Tuning follows the same form as the third stage (English-only version).

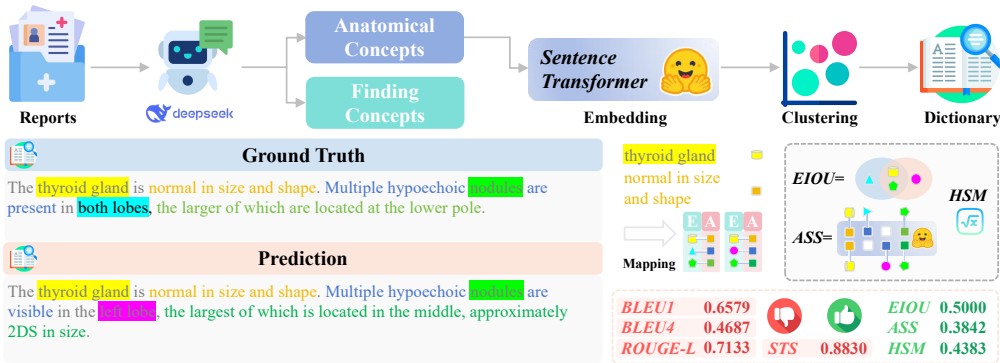

Figure 16: Workflow of anatomical and finding concepts extraction and concept-based semantic similarity metric calculation (English-only version).

| Ground Truth | SFT | AFCU SFT + CRS |
|---|---|---|
| Breast: All layers of soft tissue were clearly visualized, with no significant abnormal echogenicity in the skin or subcutaneous fat. The glands were regularly arranged bilaterally, with uniform internal echogenicity. No ductal dilatation or clear space-occupying lesions were observed, and CDFI revealed no abnormal blood flow signals. Bilateral axillary scans revealed no enlarged lymph nodes. | Scanning the area indicated by the patient on the outer side of the left breast, a relatively strong echo nodule was seen in the subcutaneous fat layer, about _3DS_ in size, with clear boundaries and regular shape. CDFI showed detectable blood flow signals. | Breast: All layers of soft tissue were clearly visualized, with no significant abnormal echogenicity in the skin or subcutaneous fat. The glands were structurally disorganized bilaterally, with heterogeneous internal echogenicity. No clear space-occupying lesions were observed, and CDFI revealed no abnormal blood flow signals. Bilateral axillary scans revealed no enlarged lymph nodes. |
| The liver is plump and has a smooth capsule. A slightly hyperechoic nodule, approximately 2DS in size, with well-defined borders and a regular shape, is visible in the left lobe. Posterior echogenicity is enhanced, and CDFI reveals detectable peripheral blood flow. The remaining parenchyma is finely echogenic, with unclear visualization of the portal venous system and enhanced echo contrast between the liver and kidneys. No intrahepatic or extrahepatic bile duct dilatation is observed. The internal diameter of the main portal vein is within the normal range. The gallbladder is normal in size and shape, with a smooth, thin wall and no obvious abnormal echogenicity within the lumen. The pancreas is plump, with slightly enhanced parenchymal echogenicity and a narrow pancreatic duct. No clear space-occupying lesion is observed within the pancreas. The spleen is normal in size and shape, with a uniform parenchymal echogenicity and no obvious space-occupying lesion within the spleen. | The liver is plump, with a smooth capsule and densely enhanced echogenicity. The portal venous system is unclear, and liver and kidney echo contrast is enhanced. The intrahepatic and extrahepatic bile ducts are not dilated. The internal diameter of the main portal vein is within the normal range. The gallbladder is normal in size and shape, with a thin, less smooth wall. Multiple hyperechoic nodules, the largest of which is approximately 2DS in diameter, are visible on the cyst wall. These nodules are silent and do not move with changes in body position. The pancreas is normal in size and shape, with a uniform echogenicity and a narrow pancreatic duct. No clear space-occupying lesions are observed. The spleen is normal in size and shape, with a uniform echogenicity and no obvious space-occupying lesions. | The liver is plump and has a smooth capsule. A cystic structure approximately 2DS in size is visible in the left lobe, with well-defined borders and increased echogenicity posteriorly. CDFI reveals no blood flow within it. The remaining parenchyma is finely echogenic, with the portal venous system less clearly visualized. Hepatic and renal echo contrast is enhanced. No intrahepatic or extrahepatic bile duct dilatation is observed. The internal diameter of the main portal vein is within the normal range. The gallbladder is normal in size and shape, with a smooth, thin wall and no obvious abnormal echogenicity within the lumen. The pancreas is normal in size and shape, with a uniform echogenicity of the parenchyma and a narrow pancreatic duct. No clear space-occupying lesion is observed within the parenchyma. The spleen is normal in size and shape, with a uniform echogenicity of the parenchyma and no obvious space-occupying lesion. |

Figure 17: Comparison of report examples generated on the mammary and liver datasets (English-only version).

We now formalize how CRS quantifies violations of these properties using information-theoretic and representation-geometric tools.

**Behavioral Rigidity as a Signature of Input Insensitivity (Information-Theoretic)** Let $\{T_i\}_{i=0}^{7}$ denote the set of reports produced from the original image ($T_0$) and seven independently occluded variants ($T_1, \ldots, T_7$). For a fixed anatomical concept $e$, define the binary indicator sequence $a_i = \mathbb{I}[e \in T_i]$ for $i = 1, \ldots, 7$. The empirical frequency

$$\mathrm{Freq}(e) = \frac{1}{7} \sum_{i=1}^{7} a_i$$

estimates the marginal probability that $e$ is emitted irrespective of visual evidence. However, frequency alone conflates systematic hallucination with stochastic noise.

To disentangle these, we model $\{a_i\}$ as i.i.d. draws from a Bernoulli distribution with parameter $p = \mathrm{Freq}(e)$. The Shannon entropy of this distribution,

$$H(e) = -p \log_2 p - (1-p) \log_2(1-p),$$

quantifies the uncertainty in the model's decision to generate $e$. Minimal entropy ($H(e) = 0$) occurs precisely when $p \in \{0, 1\}$, i.e., when the output is deterministic—either always or never generating $e$. We define stability as

$$\mathrm{Stability}(e) = 1 - H(e),$$

which maps entropy to a measure of behavioral rigidity in $[0, 1]$. The product $\mathrm{Freq}(e) \cdot \mathrm{Stability}(e)$ thus isolates concepts that are both frequently generated and insensitive to occlusion—a signature of false robustness, wherein the model exhibits unwarranted confidence due to reliance on non-visual priors.

**Content Uncertainty via Semantic Dispersion (representation-geometric)** Even when $e$ is correctly detected, clinical utility requires that its associated findings be semantically coherent across views. Let $\mathcal{F}_e^{(i)}$ denote the set of finding phrases attributed to $e$ in report $T_i$, and let $\phi_i = \text{SBERT}(\mathcal{F}_e^{(i)}) \in \mathbb{R}^d$ be their aggregated embedding (Reimers & Gurevych, 2019). Define the average cosine similarity between perturbed and reference findings as

$$s = \frac{1}{7} \sum_{i=1}^{7} \cos(\phi_i, \phi_0) = \frac{1}{7} \sum_{i=1}^{7} \frac{\phi_i^\top \phi_0}{\|\phi_i\|\|\phi_0\|}.$$

Under perfect visual grounding, $\phi_i \approx \phi_0$ for all $i$, yielding $s \to 1$. Under complete semantic randomness, $\mathbb{E}[\cos(\phi_i, \phi_0)] \to 0$ in high dimensions. Critically, the regime $s \approx 0.5$ corresponds to structured but inconsistent descriptions—indicative of unstable grounding where the model produces plausible yet mutually contradictory findings.

To emphasize this ambiguous regime, we define the Semantic Ambiguity Index (SAI) as

$$\text{SAI}(e) = \sqrt{|s(1-s)|}.$$

This function attains its maximum at $s = 0.5$ and is symmetric about this point, providing a bounded measure ($\text{SAI}(e) \in [0, 0.5]$) of semantic dispersion that penalizes both over-consistency ($s \approx 1$) and pure noise ($s \approx 0$).

**Joint Failure Detection via Multiplicative Scoring** The CRS combines the above signals multiplicatively:

$$\text{CRS}(e) = \big[\text{Freq}(e) \cdot \text{Stability}(e)\big] \cdot \text{SAI}(e).$$

This formulation ensures that high scores arise only when *both* conditions hold: The concept is persistently generated despite occlusion (high Freq · Stability), indicating visual disengagement; Its descriptive findings are semantically inconsistent (high SAI), indicating unreliable grounding.

Consequently, CRS inherently suppresses three classes of low-leverage cases: (i) rare errors (low Freq), (ii) erratic outputs (low Stability), (iii) consistently correct or consistently incorrect descriptions (low SAI).

Thus, CRS provides a theoretically grounded prioritization criterion: it identifies precisely those concepts whose correction yields maximal improvement in vision-language alignment per unit physician effort. Unlike confidence-based metrics, CRS operates solely on observable behavioral responses to perturbations, making it a more reliable proxy for visual grounding quality in safety-critical clinical settings.

## A.19 AFCU Enables Clinically Grounded and Fine-Grained Human–AI Interaction

As shown in Figure 2, a phrase refers to a short text segment obtained by splitting a report at commas or periods, a sentence refers to a longer segment split only at periods, and a report denotes the complete ground-truth report. GPT concepts are generated using the prompt: "Can you provide concise radiology descriptors for [thyroid nodules]? List in bullet points with no extra context." AFCU concepts are extracted using the prompts illustrated in Figures 8.

Regarding the results in Table 1, SFT denotes standard supervised fine-tuning using the full ground-truth report with LoRA and full visual encoder tuning. Phrase-level means feeding individual phrases (comma- or period-delimited) into the SFT-ed model for generation. Sentence-level means feeding whole sentences (period-delimited) into the SFT-ed model. Report-level means providing the entire ground-truth report as input to the SFT-ed model. GPT SFT refers to fine-tuning on the same dataset while augmenting the prompt with GPT concepts. AFCU SFT refers to fine-tuning with prompts augmented by our proposed AFCU concepts.

From Table 1, we observe that SFT + Report-level yields the best performance because the complete ground-truth report is provided as additional input—effectively giving the model the "answer." However, this setting is clinically impractical: it makes no sense to ask a radiologist to write a full report first and then collaborate with a VLM. Nonetheless, it serves as a theoretical upper bound

for human–AI collaboration. In contrast, AFCU-based interaction achieves the best performance among practically feasible approaches.

Moreover, as illustrated in Figure 2, both report-level and sentence-level inputs are too coarse-grained for interactive correction. In clinical practice, radiologists rarely need to revise an entire sentence or full report; errors are typically localized—minor omissions or inaccuracies in specific descriptions. On the other hand, phrase-level segments (split by commas) often lack semantic completeness and are thus unsuitable.

GPT concepts, while useful, tend to describe attributes in a weakly anchored manner—for example, "hyperechoic nodule" without clearly specifying which lobe (left or right), leading to ambiguity. In contrast, our AFCU framework explicitly separates: the anatomical concept (e.g., "right lobe"), and the finding concept (e.g., "hyperechoic nodule"). Only their combination yields a clear, unambiguous, atomic-level description. This design is fine-grained—enabling efficient human–AI interaction—and strongly anchored to specific anatomical locations, eliminating ambiguity.

As further supported by Table 1, our AFCU-based approach aligns most closely with real-world clinical workflows—precisely the motivation behind our original design.

## A.20 HSM ALIGNS EVALUATION WITH CLINICAL REASONING

In automatic evaluation of medical reports, existing metrics share a fundamental blind spot: they fail to model the core clinical structure of radiology reports—the binding between anatomical localization and associated findings. GREENScore (Ostmeier et al., 2024) relies on large language models to classify errors; while interpretable, it reduces evaluation to error counting and ignores contextual structure. RaTEScore (Zhao et al., 2024) and GEMA-Score (Zhang et al., 2025) introduce fine-grained entities (e.g., location, severity), yet still treat anatomy and findings as independent elements. This allows models to mask incorrect descriptions by merely mentioning the correct anatomical region—for instance, reporting "pneumothorax" instead of the true "infiltrate" in the "right lower lobe," potentially still receiving a high score—creating a serious misalignment with clinical practice.

HSM's key contribution is reframing evaluation as validation of the clinical reasoning chain. It explicitly separates two necessary conditions: (1) whether the correct anatomical regions are covered (Anatomical IoU, or AIoU), and (2) whether the description for each region is semantically accurate (Finding Semantic Similarity, or FSS). By combining them via geometric mean, HSM ensures a high score only when the model both avoids missing or hallucinating anatomy and provides accurate descriptions. Moreover, HSM does not critically depend on closed-source models—only a one-time use of an LLM is needed during dictionary initialization (which can then be reused indefinitely)—nor on complex rule systems. Instead, it relies solely on a lightweight normalization dictionary and open-source sentence embeddings, enabling an end-to-end, reproducible, and deployable evaluation framework.

## A.21 ABLATION ON CORRECTION METHODS USING ZERO-SHOT LLM

Table 18: Quantitative evaluation of report correction strategies using a zero-shot Qwen-8B on 50 mammography cases. The initial reports are generated by a US-fine-tuned Qwen-VL-3B SFT model, and then revised via either sentence-level or AFCU doctor inputs.

| SFT Report | Sentence Prompt | AFCU Prompt | origin Qwen 8B | Seed | BLEU-1 | BLEU-4 | METEOR | ROUGE-L | BLEU-1 | BLEU-4 | METEOR | ROUGE-L |
|---|---|---|---|---|---|---|---|---|---|---|---|---|
| √ | | | | - | 0.7404 | 0.5998 | 0.4270 | 0.7709 | 0.7404 | 0.5998 | 0.4270 | 0.7709 |
| √ | √ | | √ | 0 | 0.7267 | 0.5968 | 0.3970 | 0.7717 | | | | |
| √ | √ | | √ | 1 | 0.7267 | 0.5968 | 0.3970 | 0.7717 | | | | |
| √ | √ | | √ | 2 | 0.7296 | 0.5982 | 0.3973 | 0.7701 | 0.7317 | 0.6002 | 0.3988 | 0.7707 |
| √ | √ | | √ | 3 | 0.7397 | 0.6067 | 0.4017 | 0.7697 | | | | |
| √ | √ | | √ | 4 | 0.7359 | 0.6024 | 0.4008 | 0.7704 | | | | |
| √ | | √ | √ | 0 | 0.7577 | 0.6252 | 0.4348 | 0.7811 | | | | |
| √ | | √ | √ | 1 | 0.7589 | 0.6207 | 0.4308 | 0.7789 | | | | |
| √ | | √ | √ | 2 | 0.7587 | 0.6203 | 0.4303 | 0.7789 | 0.7552 | 0.6149 | 0.4285 | 0.7751 |
| √ | | √ | √ | 3 | 0.7592 | 0.6209 | 0.4308 | 0.7792 | | | | |
| √ | | √ | √ | 4 | 0.7415 | 0.5874 | 0.4157 | 0.7574 | | | | |

To isolate the effect of the correction methods themselves, we fix the initial report (generated by an ultrasound-fine-tuned Qwen-VL-3B SFT) and evaluate two prompting strategies on a zero-shot

Qwen3-8B Yang et al. (2025): (1) sentence-level instructions from (Pellegrini et al., 2025), and (2) our proposed Anatomical-Finding Concept Unit (AFCU) format. Table 18 shows that sentence-level correction degrades METEOR (from 0.4270 to 0.3988) with negligible improvement in other metrics—indicating that unstructured natural language fails to effectively guide the LLM due to referential ambiguity. For example, feedback such as "The nodule communicates with the duct..." does not specify which of multiple nodules (e.g., in the left or right breast) it refers to, leading to error-prone revisions.

In contrast, AFCU explicitly anchors each finding to a specific anatomical location (e.g., "Left breast nodule communicates with the duct; no blood flow detected"), eliminating ambiguity and consistently improving all metrics (e.g., BLEU-1 from 0.7404 to 0.7552). This improvement stems from our structured interaction design—not the model's inherent capabilities—validating AFCU as a superior correction protocol.

