# OpenReview forum: "Bidirectional Collaborative Medical Report Generation via Concept-level Interaction"
_ICLR.cc/2026/Conference — Submitted to ICLR 2026_

### Official Review · Reviewer_kJoU · 2025-10-20

**Soundness:** 3
**Presentation:** 1
**Contribution:** 2
**Rating:** 2
**Confidence:** 4

**Summary:**

The paper motivates that a new collaborative radiology reporting paradigm is required, as clinician oversight is an any case still required for AI-generated reports. They define “Report Collaboration Levels” in the current literature, arguing current works fall in level 1 and 2, which do not allow interactive/bidirectional collaboration. They propose a collaborative approach, where AI and radiologist interact on a “Anatomy-Finding-Contept Unit”, which they propose as a tuple of anatomy and finding, which they extract using DeepSeek, followed by semantic clustering using Sentence Transformer embeddings. They propose the “Concept Risk Score” as a metric to detect which of their concepts are too stable or not stable enough and therefore likely wrong and the “Holistic Semantic Match Metric” as a new report generation metric measuring IoU over the concepts they defined combined with cosine similarity of findings. The results show one round of correction improves the report generation result.

**Strengths:**

- Interaction on an abstracted level seems promising for reducing interaction overhead
- Minimal physician action leads to notable gains in performance on report generation
- Rating error risk and deciding accordingly what to let the clinician correct makes sense and can reduce workload and the proposed uncertainty score seems to work better than other methods

**Weaknesses:**

- Contribution claims: The five-fold contribution is over-claimed. I would say the main novelty lies in the concept-level interaction together with the uncertainty estimation \- the finetuning, and the metric as well as improved results should not be listed as unique contributions
- Clarity:
  - The authors define a lot of new names for rather simple concepts, making the paper hard to follow and overstating the contribution
  - Figures 2 and 3 are very unclear
  - Its not clear what table 1 shows and the results are not discussed in the result section, only shortly in the introduction
- Method:
  - The collaborative process is not really explained in the methodology, even though this is the main promise of the paper
  - It's unclear if the fine-tuning process described in section 3.2 is needed. It seems like today's LLMs would be able to adapt a report given a flagged error without fine-tuning
  - It seems like the only “interaction” is once correcting one finding that their risk score is classified as most probably wrong. That does not solve the problem of radiologists needing to double-check reports because there remains a lot of unchecked content in the report nevertheless. Why not allow correcting more than one finding if more findings are uncertain?
- Results:
  - It's unclear how the results of the proposed method are collected \- is a radiologist
  - It is not surprising that performance improves when correcting mistakes. The method is only compared to one pass report generation methods without correction \- this does not really show that the proposed way of interaction is better than other interaction modes. A comparison of basic interaction modes in both time it takes as well as reached accuracy (judged by an external reader instead of an arbitrarily defined score) would be needed in my opinion
  - It’s not clear why a new report generation metric is needed as opposed to using a similar concept as e.g. proposed in GreenScore.
  - It’s unclear how the method was evaluated / how the interaction was simulated to get the corrected report.

**Questions:**

- Why are finding concepts “clinically discriminative”? Also there ambiguity exists, e.g. “enlarged heart” vs “cardiomegaly”
- How is the theoretical upper bound determined? Why is it not 1 for a fully correct report?

---

> ### Author Response · Authors · 2025-11-24
> **Response to Reviewer kJoU (Part 1/4)**
>
> Dear Reviewer kJoU,
>
> We appreciate your constructive comments and suggestions. We remain at your disposal for any additional clarifications.
>
> **Q1:Overstated claims of contribution**
>
> **R1:** We have revised the contribution statement and now organize it into three main contributions (see Line 99-107  for details): (i) We propose, for the first time, a bidirectional collaborative MRG framework, offering a novel perspective focused on trustworthiness and reducing physicians’ workload—a specific viewpoint not previously explored in the literature. (ii) We clarify why concept-level interaction serves as the most precise atomic language for human-AI collaboration, detailing how such concepts are extracted and leveraged in a fine-tuning strategy to enable vision-language models (VLMs) to understand physicians’ instructions and generate refined reports accordingly.(iii) We introduce CRS, an uncertainty estimation method grounded in VLM behavior.
>
> The proposal of new metrics and performance improvements are regarded as supplementary contributions that provide fresh insights for medical report generation.
>
> **Q2: Define a lot of new names.**
>
> **R2:** In fact, we primarily define anatomical concepts and ​finding concepts​, and we have added a new section (Appendix A.19, Line 1572-1605) to explain the motivation and necessity behind these novel definitions.
>
> Moreover, we have revised most parts of the manuscript that could cause ambiguity and highlighted these changes to further improve readability.
>
> **Q3: Figures 2 and 3 are very unclear.**
>
> **R3:** For Figure 2, we have added a new section (Appendix A.19, Line 1593-1605) to discuss it in detail.  Both report-level and sentence-level inputs are too coarse-grained for interactive correction. In clinical practice, radiologists rarely need to revise an entire sentence or full report; errors are typically localized—minor omissions or inaccuracies in specific descriptions. On the other hand, phrase-level segments (split by commas) often lack semantic completeness and are thus unsuitable. GPT concepts, while useful, tend to describe attributes in a weakly anchored manner—for example, “hyperechoic nodule” without clearly specifying which lobe (left or right), leading to ambiguity. In contrast, our AFCU framework explicitly separates: the anatomical concept (e.g., “right lobe”), and the finding concept (e.g., “hyperechoic nodule”). Only their combination yields a clear, unambiguous, atomic-level description. This design is fine-grained—enabling efficient human–AI interaction—and strongly anchored to specific anatomical locations, eliminating ambiguity.
>
> For Figure 3, we have added explanatory text describing the three steps (Line 185-189) : (1) A fine-tuned VLM generates an initial report; (2) CRS identifies the anatomical concept with the highest error risk, and a physician provides the corrected description; (3) The VLM incorporates this feedback to produce the final revised report. And we also provide an English-only version of the image to improve readability (Figure 15).

---

> > ### Author Response · Authors · 2025-11-24
> > **Response to Reviewer kJoU (Part 2/4)**
> >
> > **Q4: Its not clear what table 1 shows and the results are not discussed in the result section, only shortly in the introduction**
> >
> > **R4:** To clarify the meaning and significance of Table 1, we have added a new subsection in the appendix A.19 (Line 1580-1606).  Regarding the results in this Table, SFT denotes standard supervised fine-tuning using the full ground-truth report with LoRA and full visual encoder tuning. Phrase-level means feeding individual phrases (comma- or period-delimited) into the SFT-ed model for generation. Sentence-level means feeding whole sentences (period-delimited) into the SFT-ed model. Report-level means providing the entire ground-truth report as input to the SFT-ed model. GPT SFT refers to fine-tuning on the same dataset while augmenting the prompt with GPT concepts. AFCU SFT refers to fine-tuning with prompts augmented by our proposed AFCU concepts. From the Table, we observe that SFT + Report-level yields the best performance because the complete ground-truth report is provided as additional input—effectively giving the model the “answer.” However, this setting is clinically impractical: it makes no sense to ask a radiologist to write a full report first and then collaborate with a VLM. Nonetheless, it serves as a theoretical upper bound for human–AI collaboration. In contrast, AFCU-based interaction achieves the best performance among practically feasible approaches.
> >
> > | Type                     | BLEU-4  | ROUGE-L | HSM     |
> > |-|-|-|-|
> > | SFT                      | 0.6341  | 0.7277  | 0.6475  |
> > | + Phrase-level           | 0.5662  | 0.6775  |  0.5984 |
> > | + Sentence-level         | 0.6481  | 0.7380  | 0.6280  |
> > | + Report-level           | 0.8669  | 0.8999  | 0.8681  |
> > | Concept-level (GPT) SFT  | 0.5653  | 0.6434  | 0.5602  |
> > | + Concept-level (GPT)    | 0.5131  | 0.6370  | 0.5470  |
> > | Concept-level (AFCU) SFT | 0.6230  | 0.6985  | 0.6282  |
> > | + Concept-level (AFCU)   | 0.6604  | 0.7413  | 0.7015  |
> >
> > **Q5: The collaborative process is not really explained in the methodology**
> >
> > **R5:** We have added a detailed description in the Method section (Line 185-188): (1) A fine-tuned VLM generates an initial report; (2) CRS identifies the anatomical concept with the highest error risk, and a physician provides the corrected description; (3) The VLM incorporates this feedback to produce the final revised report. Moreover, our method is entirely simulation-based.
> >
> > **Q6: Whether the fine-tuning process described in Section 3.2 is necessary.**
> >
> > **R6:** We have added a detailed ablation study in the Appendix A.14 (Line 1340-1358). The results show that Concept Instruction Tuning leads to substantial improvements over the base model with prompting alone (e.g., BLEU-1: 0.1043 → 0.6951), demonstrating that standard prompting is insufficient for reliable medical report generation and fine-tuning is essential to approach the upper bound.
> >
> > |                       | BLEU-1 | BLEU-4  |  METEOR |  ROUGE-L |
> > |-|-|-|-|-|
> > | Base + Prompt         | 0.1043 | 0.0301  | 0.1327  | 0.3478   |
> > | Concept Instruction Tuning + Prompt|0.6951|0.5931|0.4342|0.7078|
> > | Concept Instruction Tuning + Prompt+ CRS|0.8064 | 0.7080  | 0.4713  | 0.7883   |
> > | Upper Bound           | 0.8469 | 0.7666  | 0.5095  | 0.8300   |
> >
> > **Q7: Why not allow correcting more than one finding if more findings are uncertain?**
> >
> > **R7:** We have added new experiments, visualizations, and discussion (Sec. 4.7, Line 498-518) to explore the impact of correcting 0–4 high-risk AFCUs on report refinement. The results show that iteratively correcting multiple high-risk AFCUs further improves model performance. Additionally, as this is our initial exploration and no established baselines exist for such multi-correction interaction, we chose to start by studying the single-correction collaborative setting. Future work will focus on enhancing the performance and efficiency of multi-correction collaborative paradigms.
> >
> > **Q8: It's unclear how the results of the proposed method are collected - is a radiologist**
> >
> > **R8:** We collect results using a simulation-based approach. Specifically, since we already have the ground-truth reports, we first decompose them into structured concepts as defined in our framework: anatomical concepts (representing anatomical structures) and finding concepts (descriptions associated with each anatomical structure; one anatomical concept may correspond to multiple finding concepts). We use an LLM with carefully designed prompts (Figure 8 and 9) to extract these concepts from the ground-truth reports. During simulated interaction, this enables the system to identify precisely which anatomical concept needs correction, and the simulated radiologist then provides the corresponding corrected finding concept. (This is also why we introduced several new conceptual definitions—not to add unnecessary complexity, but because they are essential for enabling this simulation and the proposed collaborative paradigm.)

---

> > > ### Author Response · Authors · 2025-11-24
> > > **Response to Reviewer kJoU (Part 3/4)**
> > >
> > > **Q9: (1) The method is only compared to one pass report generation methods without correction (2) A comparison of basic interaction modes in both time it takes as well as reached accuracy**
> > >
> > > **R9:** (1) To the best of our knowledge, we are the first to propose a correction-based human–AI collaboration framework for medical report generation. There are no existing comparable methods for this specific interaction paradigm. As for conventional report generation models (e.g., CNN- or ViT-based architectures), they are inherently unable to accept additional textual feedback for iterative refinement—such modification-capable interaction mechanisms simply do not exist in those frameworks.
> > >
> > > Currently, only vision-language models (VLMs) support iterative refinement via textual prompts. We have evaluated multiple VLMs under identical prompting strategies, and all yield significantly worse performance than our CRS-guided approach (see Appendix A.10, Line 1222-1241). Moreover, the optimal form of human feedback—whether a full report, a sentence, or a concise yet informative cue—remains an open research question. Our work is the first to systematically investigate and validate an effective interaction protocol, which we consider a key contribution.
> > >
> > > | Model              | BLEU-1  | BLEU-4  | METEOR  | ROUGE-L  | Precision  | Recall  | F1 Score  | AIOU    | FSS     | HSM    |
> > > |--------------------|---------|---------|---------|----------|------------|---------|-----------|---------|---------|--------|
> > > | Lingshu-32B        | 0.2207  | 0.0153  | 0.1348  | 0.1573   | 0.5537     | 0.2375  | 0.3123    | 0.2049  | 0.0488  | 0.0968 |
> > > | Qwen2.5-VL-3B      | 0.0846  | 0.0055  | 0.0912  | 0.0957   | 0.2484     | 0.1593  | 0.1858    | 0.1187  | 0.0246  | 0.0507 |
> > > | Medgemma-4b        | 0.1059  | 0.0075  | 0.1148  | 0.1123   | 0.2412     | 0.1450  | 0.1746    | 0.1125  | 0.0228  | 0.0480 |
> > > | Qwen2.5-VL-3B+SFT  | 0.6220  | 0.4367  | 0.3284  | 0.6042   | 0.8541     | 0.8215  | 0.8260    | 0.7281  | 0.5622  | 0.6351 |
> > > | Medgemma-4b+SFT    | 0.6248  | 0.4350  | 0.3318  | 0.6039   | 0.8635     | 0.8219  | 0.8311    | 0.7378  | 0.5712  | 0.6445 |
> > > | Medgemma-4b+Ours   | 0.6642  | 0.4609  | 0.3558  | 0.6107   | 0.8879     | 0.8676  | 0.8689    | 0.7877  | 0.5916  | 0.6777 |
> > >
> > > (2) Regarding evaluation metrics, we do include assessments by external clinical experts, which have been shown to be highly correlated with our proposed HSM score (Figure 5a). While having radiologists evaluate every single report would be ideal, it is prohibitively expensive and impractical at scale.
> > >
> > > Concerning time cost, we have added a new experiment in Appendix A.16 (Line1388-1403) demonstrating that Our method—AI-generated reports with CRS-based checking followed by physician editing of one AFCU—reduced the average annotation time by 14.34 seconds compared to conventional AI-assisted editing without CRS, demonstrating a measurable reduction in physician workload.
> > >
> > > | Human-Generated  | AI-Generated  | Human Checking  | CRS  | Human Editing  | Time (s) |
> > > |-|-|-|-|-|-|
> > > | √                |               |                 |      |                | 205.95   |
> > > |                  | √             |                 |      |                | 15.16    |
> > > |                  | √             | √               |      | √              | 46.10    |
> > > |                  | √             |                 | √    | √              | 31.76    |
> > >
> > > **Q10: It’s not clear why a new report generation metric is needed as opposed to using a similar concept as e.g. proposed in GreenScore.**
> > >
> > > **R10:** We have added a new subsection (Appendix A.20, Line 1607-1627), which directly compares HSM to existing semantic metrics such as RaTEScore [1], GREENScore [2], and GEMA-Score [3]. We explain how these methods—while valuable—still treat anatomical locations and findings as independent components, leading to misalignment with clinical reasoning. In contrast, HSM explicitly models their binding through Anatomical IoU (AIoU) and Finding Semantic Similarity (FSS), ensuring that both correct localization and accurate description are required for a high score. This design better reflects real-world radiology assessment and avoids inflated scores resulting from partial correctness.

---

> > > > ### Author Response · Authors · 2025-11-24
> > > > **Response to Reviewer kJoU (Part 4/4)**
> > > >
> > > > **Q11: It’s unclear how the method was evaluated / how the interaction was simulated to get the corrected report.**
> > > >
> > > > **R11:** First, Figure 3 (Line 229-247) illustrates the overall pipeline. To improve readability, we provide a fully English version of the figure along with the procedural description in Line Figure 15 (Line 1429-1447).
> > > >
> > > > Second, our simulation relies on a pre-constructed dictionary that maps each ground-truth report to its anatomical concepts and corresponding finding concepts. After generating a report, we use this dictionary to extract the anatomical and finding concepts from the generated text. CRS then estimates the uncertainty of each anatomical concept, and the (simulated) radiologist provides the corrected finding concept associated with the highest-risk anatomical concept. A concrete example is shown in Figure 3.
> > > >
> > > > **Q12: Why are finding concepts “clinically discriminative”?**
> > > >
> > > > **R12:** Finding concepts are determined by the perceptual similarity threshold described in Line 190-229, which is controlled—and, when necessary, manually refined—by researchers with medical expertise. Therefore, the "clinically discriminative" nature of these concepts depends on both the concept extraction process and the medical knowledge and judgment of the annotators.
> > > >
> > > > **Q13: How is the theoretical upper bound determined? Why is it not 1 for a fully correct report?**
> > > >
> > > > **R13:** First, UB denotes the theoretical upper bound under Concept Instruction Fine-tuning with intervention—specifically, the result obtained by feeding the complete ground-truth report as a prompt into the SFT-ed VLM. We have added this clarification in Appendix A.19, Line 1587-1592.
> > > >
> > > > Second, in Appendix A.19, Line 1587-1592, we explain why providing the full ground-truth report as input to the VLM does not exactly reproduce the original report: this is because VLMs exhibit inherent stochasticity—even with identical input prompts, their outputs can vary due to random sampling during generation.
> > > >
> > > >
> > > > **Reference**
> > > >
> > > > [1] Ratescore: A metric for radiology report generation, arXiv, 2024
> > > >
> > > > [2] Green: Generative radiology report evaluation and error notation, EMNLP, 2024
> > > >
> > > > [3] GEMA-Score: Granular Explainable Multi-Agent Score for Radiology Report Evaluation, arXiv, 2025

---

> > > > > ### Comment · Reviewer_kJoU · 2025-11-25
> > > > >
> > > > > Thank you for your extensive rebuttal. While I still think the paper lacks overall in clarity, I appreciate the additional descriptions in regard to Q3, Q4, Q5 and Q8. I also appreciate the additional results provided regarding Q7 (correcting multiple findings), they add value to the paper.
> > > > >
> > > > > Regarding Q6 and Q9, I think you have miss-understood my main concern. For both these new ablations, it seems VLMs that are not trained on on the US dataset are now given as comparison - as they will have a bad result already for the initial report, their bad metrics after correction are not surprising. My main question is indeed only about the correction part. A fitting ablation for this would e.g. be to use the report of a US-fine-tuned model e.g. Qwen 3b SFT from Table 2 and give this report together with a correction prompt to a zero-shot LLM e.g. again Qwen (as the fine-tuned model probably lost conversational abilities during fine-tuning) and report the results of the corrected report. It would be interesting to see this for both a "trivial" correction prompt, e.g. using full sentences and the proposed concept-level interaction. This form of basic VLM/LLM correction has also been proposed before e.g. in [1], so the claim that this does not exist is not correct.
> > > > >
> > > > > [1] Pellegrini, C., Özsoy, E., Busam, B., Wiestler, B., Navab, N., & Keicher, M. (2025). Radialog: Large vision-language models for x-ray reporting and dialog-driven assistance. In Medical Imaging with Deep Learning.
> > > > >
> > > > > Regarding contibution, I think the contribution is still overclaimed. I remain with my opinion that there is one main contribution, which is the concept-level interaction *in combination* with the uncertainty estimation with the main result being the better selection of risk findings compared to e.g. random correction (Table 3). If the method proves to be significantly better in correcting based on the proposed AFCUs in above described experiments, this would also be a relevant result. Therefore it is not enough in my opinion to slighly rewrite the contribution paragraph, but instead the whole paper would need to be reframed both in terms of contribution and overall clarity.
> > > > >
> > > > > For these reasons, my recommendation to reject the paper in its current form remains.

---

> > > > > > ### Author Response · Authors · 2025-12-03
> > > > > > **Response to Reviewer kJoU (Part 1/2)**
> > > > > >
> > > > > > Dear Reviewer kJoU,
> > > > > >
> > > > > > We sincerely appreciate your feedback and guidance. Importantly, our approach is not a conventional correction method—unlike RaDialog, which requires physicians to fully review both images and reports before issuing edits. Instead, we use the VLM’s local uncertainty estimates to guide physicians to inspect and revise only specific anatomical descriptions, significantly reducing their workload. This human-AI collaboration paradigm is, to our knowledge, unexplored in prior work. We also provide experiments showing that our method improves correction performance.
> > > > > >
> > > > > > **Q1: New experiments (SFT 3B generates the initial report + different prompts + original Qwen-8B); comparison with Radialog's correction approach.**
> > > > > >
> > > > > > **R1:** To rigorously isolate the effect of the correction methods, we have added new experiments and clarifications (Appendix A.21). We fix the initial report to be generated by an ultrasound-domain fine-tuned VLM (Qwen-VL-3B SFT) and apply two prompting strategies to a zero-shot LLM (Qwen-8B): (1) a sentence-level instruction derived from Radialog via DeepSeek-based extraction, and (2) our anatomical-finding-concept-unit (AFCU) format. As shown in Table, the sentence-level approach consistently degrades METEOR (from 0.4270 to 0.3988) with only marginal changes in other metrics, indicating that even factually correct natural-language feedback fails to guide precise revision due to lack of structural grounding. In contrast, AFCU-based prompting improves all metrics (e.g., BLEU-1 from 0.7404 to 0.7552), demonstrating that explicit concept-level anchoring enables more faithful correction.
> > > > > >
> > > > > > | Methods    |                 |             |                 | Seed  | Metric |        |        |         | Average |        |        |          |
> > > > > > |-|-|-|-|-|-|-|-|-|-|-|-|-|
> > > > > > | SFT Report | Sentence Prompt | AFCU Prompt | origin Qwen 8B  |       | BLEU-1 | BLEU-4 | METEOR | ROUGE-L | BLEU-1  | BLEU-4 | METEOR | ROUGE-L  |
> > > > > > | √          |                 |             |                 | -     | 0.7404 | 0.5998 | 0.4270 | 0.7709  | 0.7404  | 0.5998 | 0.4270 | 0.7709   |
> > > > > > | √          | √               |             | √               | 0     | 0.7267 | 0.5968 | 0.3970 | 0.7717  | 0.7317  | 0.6002 | 0.3988 | 0.7707   |
> > > > > > | √          | √               |             | √               | 1     | 0.7267 | 0.5968 | 0.3970 | 0.7717  |         |        |        |          |
> > > > > > | √          | √               |             | √               | 2     | 0.7296 | 0.5982 | 0.3973 | 0.7701  |         |        |        |          |
> > > > > > | √          | √               |             | √               | 3     | 0.7397 | 0.6067 | 0.4017 | 0.7697  |         |        |        |          |
> > > > > > | √          | √               |             | √               | 4     | 0.7359 | 0.6024 | 0.4008 | 0.7704  |         |        |        |          |
> > > > > > | √          |                 | √           | √               | 0     | 0.7577 | 0.6252 | 0.4348 | 0.7811  | 0.7552  | 0.6149 | 0.4285 | 0.7751   |
> > > > > > | √          |                 | √           | √               | 1     | 0.7589 | 0.6207 | 0.4308 | 0.7789  |         |        |        |          |
> > > > > > | √          |                 | √           | √               | 2     | 0.7587 | 0.6203 | 0.4303 | 0.7789  |         |        |        |          |
> > > > > > | √          |                 | √           | √               | 3     | 0.7592 | 0.6209 | 0.4308 | 0.7792  |         |        |        |          |
> > > > > > | √          |                 | √           | √               | 4     | 0.7415 | 0.5874 | 0.4157 | 0.7574  |         |        |        |          |
> > > > > >
> > > > > > The limitation of sentence-level feedback is referential ambiguity. For example, a statement like “The nodule communicates with the duct but shows no blood flow” does not specify which of multiple nodules (e.g., left breast, right breast, axillae) it refers to—creating a many-to-one mapping between text and anatomy. AFCU resolves this by explicitly binding findings to structures: “Exist left breast nodule, appears communicating with duct, no blood flow detected.”

---

> > > > > > > ### Author Response · Authors · 2025-12-03
> > > > > > > **Response to Reviewer kJoU (Part 2/2)**
> > > > > > >
> > > > > > > **Q2: Overclaimed contribution.**
> > > > > > >
> > > > > > > **R2:** We have now added experiments (Appendix A.21) that demonstrate the effectiveness of our correction approach. As you noted, such a result would constitute a meaningful contribution, which our experiments now substantiate.
> > > > > > >
> > > > > > > Moreover, our approach is not merely a correction method (like Radialog[1] you have cited). Instead, it actively evaluates the local uncertainty of the text generated by the Vision-Language Model and specifically instructs physicians to revise only the corresponding localized segments of the report. This is fundamentally different from Radialog, which requires physicians to comprehensively review the entire medical report alongside the full medical image. In our method, physicians examine only specific local image regions and their associated textual descriptions—guided by the VLM’s assessment of uncertainty regarding the descriptions of different anatomical structures. This represents a more advanced paradigm than RaDialog. In fact, we clearly define the three RCL levels in Figure 1 and Line 48-57 of the Introduction, where Radialog falls under RCL-1, while our work presents the first RCL-3 framework.
> > > > > > >
> > > > > > > To the best of our knowledge, no prior work has explored this interaction mode. Accordingly, our concept-level investigations—centered on defining and formalizing this novel paradigm—constitute unique contributions of our work. Moreover, we do not believe that articulating multiple contributions necessarily amounts to overclaiming. Rather, we aim to clearly articulate how we progressively understood the problem and developed corresponding solutions, and we hope that summarizing the key methodological insights gained through our exploration will provide meaningful inspiration to readers.
> > > > > > >
> > > > > > > Best regards,
> > > > > > >
> > > > > > > Authors

---

### Official Review · Reviewer_yPVd · 2025-10-23

**Soundness:** 3
**Presentation:** 1
**Contribution:** 3
**Rating:** 2
**Confidence:** 4

**Summary:**

This paper presents a bidirectional collaborative framework for medical report generation that enhances model trustworthiness while reducing physicians’ workload. It leverages information bottleneck theory for concept-level feedback refinement and introduces the Concept Risk Score to prioritize high-impact diagnostic corrections. Additionally, it establishes concept-grounded semantic metrics and demonstrates a 9.13% improvement in report quality with minimal physician intervention.

**Strengths:**

- The move to concept-level feedback refinement is original and relevant for real-world applications.
- The use of semantic similarity to go beyond simple overlap metrics is well motivated and adds depth. Also comparison with other metrics is beneficial.
- The correlation analysis between the proposed metrics and human judgments / LLM evaluations is a strong plus and a meaningful evaluation.
- Report a quantifiable improvement (9.13%).

**Weaknesses:**

* The authors claim a five-fold contribution, but the paper really delivers two main contributions: the bidirectional collaborative interaction, uncertainty estimation and maybe the new metric.
* Authors redefine existing definitions, making it hard to read the paper and confusing.
* The presentation is uneven: mixing languages in figures, and a methodology section that interleaves high‐level design with implementation details and discussions, hurts readability.
* It would be nice to see the proper evaluation of each part of the method, to understand where exactly the contribution is and what helped to improve the performance.
* The paper has to be rewritten.
* There are some interesting results in the tables, but they are not properly discussed or explained.

**Questions:**

If the metric goes as a contribution, it would be worth comparing it to other semantic metrics like RaTEScore, GreenScore, GEMA‑Score and etc.

The authors introduce a collaborative interaction mechanism, yet they do not compare it to other methods under the same setup. Since giving one method the possibility to correct its mistakes makes it obvious that it will perform better, the more meaningful question is: if other competing methods were given the same correction-capability setup, would the proposed method still outperform them?

---

> ### Author Response · Authors · 2025-11-24
> **Response to Reviewer yPVd (Part 1/3)**
>
> Dear Reviewer yPVd,
>
> Thanks a lot for your valuable time and constructive comments. Should you have any follow-up questions, we would be honored to respond.
>
> **Q1:Overstated claims of contribution**
>
> **R1:** We have revised the contribution statement and now organize it into three main contributions (see Line 99-107  for details): (i) We propose, for the first time, a bidirectional collaborative MRG framework, offering a novel perspective focused on trustworthiness and reducing physicians’ workload—a specific viewpoint not previously explored in the literature. (ii) We clarify why concept-level interaction serves as the most precise atomic language for human-AI collaboration, detailing how such concepts are extracted and leveraged in a fine-tuning strategy to enable vision-language models (VLMs) to understand physicians’ instructions and generate refined reports accordingly.(iii) We introduce CRS, an uncertainty estimation method grounded in VLM behavior.
>
> The proposal of new metrics and performance improvements are regarded as supplementary contributions that provide fresh insights for medical report generation.
>
> **Q2: Redefine existing definitions.**
>
> **R2:** We have revised most parts of the manuscript that could cause ambiguity and highlighted these changes to further improve readability.
>
> **Q3: mixing languages in figures, and a methodology section that interleaves high‐level design with implementation details and discussions, hurts readability.**
>
> **R3:** Regarding the issue of mixing languages, we have provided English-only versions of all relevant figures in Appendix A.17 (Lines 1404–1412) to further enhance accessibility.
>
> However, we emphasize that all our experiments are based on a Chinese ultrasound report dataset; therefore, retaining the original Chinese text is essential for academic rigor, following the practice of KMVE [1]. Moreover, every Chinese segment is accompanied by an English translation to improve readability, as noted in Line 374-375.
>
> Regarding your comment on the method section, we structured it with a high-level design followed by implementation details to help readers better understand the motivation behind our approach. As for the discussion on CRS, we have adopted your suggestion and moved it to Section 4.3 (Lines 426–452) in the Experiments section to improve readability. Furthermore, we have added a theoretical analysis of CRS in Appendix A.18 (Lines 1449–1570) to further facilitate readers’ understanding of our method.
>
> **Q4: evaluation of each part of the method, to understand where exactly the contribution is and what helped to improve the performance.**
>
> **R4:** We have added a detailed ablation study in the Appendix A.14 (Line 1340-1356). As shown in Table 15, the base model with only intervention prompts performs poorly, confirming that prompt-following ability is not inherent to the VLM. Concept Instruction Tuning provides the main performance gain by aligning the model with clinically structured concepts, and CRS further refines outputs through uncertainty-aware intervention. Together, they enable our framework to approach the theoretical upper bound, clearly attributing the improvement to these two components.
>
> |                       | BLEU-1 | BLEU-4  |  METEOR |  ROUGE-L |
> |-|-|-|-|-|
> | Base + Prompt         | 0.1043 | 0.0301  | 0.1327  | 0.3478   |
> | Concept Instruction Tuning + Prompt|0.6951|0.5931|0.4342|0.7078|
> | Concept Instruction Tuning + Prompt+ CRS|0.8064 | 0.7080  | 0.4713  | 0.7883   |
> | Upper Bound           | 0.8469 | 0.7666  | 0.5095  | 0.8300   |

---

> > ### Author Response · Authors · 2025-11-24
> > **Response to Reviewer yPVd (Part 2/3)**
> >
> > **Q5: The paper has to be rewritten.**
> >
> > **R5:** We have thoroughly rewritten and substantially expanded many unclear descriptions; please refer to the highlighted portions in our revised manuscript.
> >
> > **Q6: There are some interesting results in the tables, but they are not properly discussed or explained.**
> >
> > **R6:** We have added discussion of interesting results in the existing tables—for example, on certain smaller datasets, the Qwen2.5-VL-3B model outperforms the 7B variant—likely because the larger model’s higher capacity leads to overfitting when training data is limited. We have added a clarification to this effect in the revised manuscript `(Line 421-424). In future work, we plan to incorporate larger-scale instruction-tuning datasets to better harness the potential of bigger models.
> >
> > **Q7: If the metric goes as a contribution, it would be worth comparing it to other semantic metrics like RaTEScore, GreenScore, GEMA‑Score and etc.**
> >
> > **R7:** We have clarified that the HSM metric is not the primary contribution of this work but rather an additional exploration (Line 99-107).
> >
> > Nevertheless, we have added a new subsection (Appendix A.20, Line 1607-1627), which directly compares HSM to existing semantic metrics such as RaTEScore [2], GREENScore [3], and GEMA-Score [4]. We explain how these methods—while valuable—still treat anatomical locations and findings as independent components, leading to misalignment with clinical reasoning. In contrast, HSM explicitly models their binding through Anatomical IoU (AIoU) and Finding Semantic Similarity (FSS), ensuring that both correct localization and accurate description are required for a high score. This design better reflects real-world radiology assessment and avoids inflated scores resulting from partial correctness.

---

> > > ### Author Response · Authors · 2025-11-24
> > > **Response to Reviewer yPVd (Part 3/3)**
> > >
> > > **Q8：The authors introduce a collaborative interaction mechanism, yet they do not compare it to other methods under the same setup. Since giving one method the possibility to correct its mistakes makes it obvious that it will perform better, the more meaningful question is: if other competing methods were given the same correction-capability setup, would the proposed method still outperform them?**
> > >
> > > **R8:**(1) To the best of our knowledge, we are the first to propose a human-AI collaborative framework that enables targeted revision and completion of medical reports at the anatomical concept level. Consequently, there are no existing methods under the same interactive setup that can serve as direct baselines for comparison.
> > >
> > > (2) While it is true that physician-in-the-loop correction generally improves report quality, prior work has not addressed what should be corrected to maximize efficiency. As noted in our paper, current AI-generated reports still require physicians to perform full manual review. In contrast, our approach introduces a novel, principled framework—Concept Risk Score (CRS)—that reliably identifies the most critical regions for intervention. This shifts the paradigm from exhaustive verification to focused, high-impact feedback, offering strong potential for real-world clinical deployment. We have added a new experiment that quantifies the time savings through a human-in-the-loop evaluation. In Appendix A.16 (Line1388-1403), we report results from an experiment using 15 samples from the Mammary dataset, comparing four annotation settings. Our method—AI-generated reports with CRS-based checking followed by physician editing of one AFCU—reduced the average annotation time by 14.34 seconds compared to conventional AI-assisted editing without CRS, demonstrating a measurable reduction in physician workload.
> > >
> > > | Human-Generated  | AI-Generated  | Human Checking  | CRS  | Human Editing  | Time (s) |
> > > |-|-|-|-|-|-|
> > > | √                |               |                 |      |                | 205.95   |
> > > |                  | √             |                 |      |                | 15.16    |
> > > |                  | √             | √               |      | √              | 46.10    |
> > > |                  | √             |                 | √    | √              | 31.76    |
> > >
> > > (3) Currently, only vision-language models (VLMs) support iterative refinement via textual prompts. We have evaluated multiple VLMs under identical prompting strategies, and all yield significantly worse performance than our CRS-guided approach (see Appendix A.10, Line 1222-1241). Moreover, the optimal form of human feedback—whether a full report, a sentence, or a concise yet informative cue—remains an open research question. Our work is the first to systematically investigate and validate an effective interaction protocol, which we consider a key contribution.
> > >
> > > | Model              | BLEU-1  | BLEU-4  | METEOR  | ROUGE-L  | Precision  | Recall  | F1 Score  | AIOU    | FSS     | HSM    |
> > > |-|-|-|-|-|-|-|-|-|-|-|
> > > | Lingshu-32B        | 0.2207  | 0.0153  | 0.1348  | 0.1573   | 0.5537     | 0.2375  | 0.3123    | 0.2049  | 0.0488  | 0.0968 |
> > > | Qwen2.5-VL-3B      | 0.0846  | 0.0055  | 0.0912  | 0.0957   | 0.2484     | 0.1593  | 0.1858    | 0.1187  | 0.0246  | 0.0507 |
> > > | Medgemma-4b        | 0.1059  | 0.0075  | 0.1148  | 0.1123   | 0.2412     | 0.1450  | 0.1746    | 0.1125  | 0.0228  | 0.0480 |
> > > | Qwen2.5-VL-3B+SFT  | 0.6220  | 0.4367  | 0.3284  | 0.6042   | 0.8541     | 0.8215  | 0.8260    | 0.7281  | 0.5622  | 0.6351 |
> > > | Medgemma-4b+SFT    | 0.6248  | 0.4350  | 0.3318  | 0.6039   | 0.8635     | 0.8219  | 0.8311    | 0.7378  | 0.5712  | 0.6445 |
> > > | Medgemma-4b+Ours   | 0.6642  | 0.4609  | 0.3558  | 0.6107   | 0.8879     | 0.8676  | 0.8689    | 0.7877  | 0.5916  | 0.6777 |
> > >
> > > **Reference**
> > >
> > > [1] Ultrasound report generation with cross-modality feature alignment via unsupervised guidance, TMI, 2024
> > >
> > > [2] Ratescore: A metric for radiology report generation, arXiv, 2024
> > >
> > > [3] Green: Generative radiology report evaluation and error notation, EMNLP, 2024
> > >
> > > [4] GEMA-Score: Granular Explainable Multi-Agent Score for Radiology Report Evaluation, arXiv, 2025

---

> > > > ### Comment · Reviewer_yPVd · 2025-11-26
> > > >
> > > > Dear authors,
> > > >
> > > > Thanks a lot for the answers.  Even though some of my concerns addressed. But the structure of the paper is still quite confusing, and checking the updated pdf I don’t see any part really being rewritten. Even though you added in appendix interesting information.  But my main concerns regarding the  collaborative interaction mechanism is still not addressed. Even though there are as authors claim no existing methods doing that, why not give other models possibility to correct themselves and see the results. It would strength your method results as well. So there is still no experiment isolating the effect of CRS itself across baseline models

---

> ### Author Response · Authors · 2025-12-03
> **Response to Reviewer yPVd (Part 1/2)**
>
> Dear Reviewer yPVd,
>
> We sincerely appreciate your initial score adjustment and valuable feedback. Following your suggestions, we have added a new experimental evaluation of the collaborative interaction mechanism. Notably, we emphasize that our core contribution is not a conventional correction method—unlike approaches requiring physicians to fully review images and reports before editing. Instead, we use the VLM’s local uncertainty estimates to guide physicians to revise only specific anatomical descriptions, significantly reducing their workload. To our knowledge, this human–AI collaboration paradigm is unexplored in prior work.
>
> **Q1: The collaborative interaction mechanism is still not addressed.**
>
> **R1:** The core of the AFCU correction mechanism lies in its fine-grained, strongly anchored binding of textual descriptions to specific anatomical structures. We have added new experiments and clarifications (Appendix A.21) to further substantiate this design.
>
> To rigorously isolate the effect of the CRS and investigate the impact of the correction component, we fix the initial report to be generated by an ultrasound-domain fine-tuned VLM (Qwen-VL-3B SFT) and apply two prompting strategies to a zero-shot LLM (Qwen-8B): (1) a sentence-level instruction derived from Radialog via DeepSeek-based extraction, and (2) our anatomical-finding-concept-unit (AFCU) format. As shown in Table, the sentence-level approach consistently degrades METEOR (from 0.4270 to 0.3988) with only marginal changes in other metrics, indicating that even factually correct natural-language feedback fails to guide precise revision due to lack of structural grounding (the reason Radialog can perform corrections in its original paper is that it underwent supervised fine-tuning). In contrast, AFCU-based prompting improves all metrics (e.g., BLEU-1 from 0.7404 to 0.7552), demonstrating that explicit concept-level anchoring enables more faithful correction.
>
> | Methods    |                 |             |                 | Seed  | Metric |        |        |         | Average |        |        |          |
> |-|-|-|-|-|-|-|-|-|-|-|-|-|
> | SFT Report | Sentence Prompt | AFCU Prompt | origin Qwen 8B  |       | BLEU-1 | BLEU-4 | METEOR | ROUGE-L | BLEU-1  | BLEU-4 | METEOR | ROUGE-L  |
> | √          |                 |             |                 | -     | 0.7404 | 0.5998 | 0.4270 | 0.7709  | 0.7404  | 0.5998 | 0.4270 | 0.7709   |
> | √          | √               |             | √               | 0     | 0.7267 | 0.5968 | 0.3970 | 0.7717  | 0.7317  | 0.6002 | 0.3988 | 0.7707   |
> | √          | √               |             | √               | 1     | 0.7267 | 0.5968 | 0.3970 | 0.7717  |         |        |        |          |
> | √          | √               |             | √               | 2     | 0.7296 | 0.5982 | 0.3973 | 0.7701  |         |        |        |          |
> | √          | √               |             | √               | 3     | 0.7397 | 0.6067 | 0.4017 | 0.7697  |         |        |        |          |
> | √          | √               |             | √               | 4     | 0.7359 | 0.6024 | 0.4008 | 0.7704  |         |        |        |          |
> | √          |                 | √           | √               | 0     | 0.7577 | 0.6252 | 0.4348 | 0.7811  | 0.7552  | 0.6149 | 0.4285 | 0.7751   |
> | √          |                 | √           | √               | 1     | 0.7589 | 0.6207 | 0.4308 | 0.7789  |         |        |        |          |
> | √          |                 | √           | √               | 2     | 0.7587 | 0.6203 | 0.4303 | 0.7789  |         |        |        |          |
> | √          |                 | √           | √               | 3     | 0.7592 | 0.6209 | 0.4308 | 0.7792  |         |        |        |          |
> | √          |                 | √           | √               | 4     | 0.7415 | 0.5874 | 0.4157 | 0.7574  |         |        |        |          |
>
> The limitation of sentence-level feedback is referential ambiguity. For example, a statement like “The nodule communicates with the duct but shows no blood flow” does not specify which of multiple nodules (e.g., left breast, right breast, axillae) it refers to—creating a many-to-one mapping between text and anatomy. AFCU resolves this by explicitly binding findings to structures: “Exist left breast nodule, appears communicating with duct, no blood flow detected.”

---

> > ### Author Response · Authors · 2025-12-03
> > **Response to Reviewer yPVd (Part 2/2)**
> >
> > **Q2: why not give other models possibility to correct themselves and see the results.**
> >
> > **R2:** In fact, our method has demonstrated effectiveness not only on Qwen-VL-3B and Qwen-VL-7B as shown in Table 2 (Line 380-416), but we have also included Appendix A.10 ("Comparison with Other Medical VLMs"), where Table 11 (Line 1242-1250) further demonstrates its efficacy on MedGemma-4B. Based on extensive experiments across four organs, we can reasonably infer that our approach is likely effective for the majority of vision-language models.
> >
> > **Q3: Rewrite the paper.**
> >
> > **R3:** We have added a description of the human-AI collaboration workflow in Lines 185–188 of the Method section, and Figure 3 provides a schematic illustration of the specific step in this collaborative process. To further improve readability, we have also included a fully English version of this workflow in the Appendix A.17. Moreover, we have added extensive supplementary material to clarify any potentially ambiguous points (Appendices A.13–A.21). Reviewers `tMhq, SJet, and TNks` have all noted that our paper features a clear and easy-to-follow presentation.
> >
> > Best regards,
> >
> > Authors

---

### Official Review · Reviewer_TNks · 2025-10-28

**Soundness:** 2
**Presentation:** 2
**Contribution:** 2
**Rating:** 4
**Confidence:** 4

**Summary:**

This paper proposes a new interactive medical report generation method. Intuitively, authors reduce physician workload in medical report generation by identifying areas of uncertainty in images. To some extent, this is reasonable.

**Strengths:**

1. The writing of this paper is good and easy to follow.

2. This paper has a certain degree of innovation, introducing the Concept Risk Score (CRS) to help models to capture the unstable regions of images.

**Weaknesses:**

1. Figures 2, 3, 6, 9, 10, and 12 contain a lot of non-English content, which is not friendly to reviewers.

2. I suggest that the authors can add relevant data on clinical efficacy (CE) metrics in Table 1, 2, 3, 10, 11, 12 and 13, as the CE metrics of the medical report generation comes first.

3. Most of the comparison algorithms are too old, making it difficult to demonstrate the effective of the method proposed in this paper. I suggest the authors add two more algorithms from the last two years for comparison.

4. The authors proposed the CRS indicator, but its reliability has not been verified. It is recommended that the authors provide a theoretical analysis of CRS.

5. The LLM, acting as a trainee, points out uncertain areas based on its own capabilities. However, the LLM itself lacks the ability to reason [1,2], making it difficult to pinpoint the few truly difficult areas and tends to point out the majority of uncertain areas. This does not reduce the workload of physicians.

[1] Are Large Language Models Really Good Logical Reasoners? A Comprehensive Evaluation and Beyond

[2] Understanding social reasoning in language models with language models

**Questions:**

Please see the weaknesses.

---

> ### Author Response · Authors · 2025-11-24
> **Response to Reviewer TNks**
>
> Dear Reviewer TNks,
>
> Thank you very much for providing valuable feedback. We are happy to engage with you for any follow-up questions.
>
> **Q1: Many figures contain non-English text, reducing reviewer accessibility.**
>
> **R1:** To further enhance accessibility, we provide English-only versions of all relevant figures in Appendix A.17 (Line1404-1412).
>
> However, we emphasize that all our experiments are based on a Chinese ultrasound report dataset; therefore, retaining the original Chinese text is essential for academic rigor, following the practice of KMVE [1]. Moreover, every Chinese segment is accompanied by an English translation to improve readability, as noted in Line 374-375.
>
> **Q2&3: Add CE metrics and the latest comparison methods.**
>
> **R2&3:** We fully agree with the reviewer that clinical efficacy (CE) is more important than textual similarity. However, our ultrasound reports do not contain disease diagnosis labels, and unlike X-ray reports—which can be labeled using CheXbert [2] to compute CE metrics—there is currently no equivalent model for ultrasound reports. Moreover, one of the current best-performing approaches on ultrasound report generation is fine-tuning QwenVL, as described in [3].
>
> To address this limitation, we conduct new experiments (Appendix A.15 Line1359-1383) on the English-language IU-Xray dataset [4], where CheXbert [2] enables reliable CE evaluation. We report CE metrics and compare against the latest methods. The results demonstrate that our method substantially improves the clinical efficacy of generated reports.
>
> |           | Year                   | METEOR |  Precision |  Recall |  F1    |
> |-|-|-|-|-|-|
> | R2Gen     |  ACL 2020              |  0.128 |  0.151     |  0.145  |  0.145 |
> | CVT2Dis   |  Artif.Intell.Med 2022 |  0.147 |  0.174     |  0.172  |  0.168 |
> | M2KT      |  MIA 2023              |  0.153 |  0.153     |  0.145  |  0.145 |
> | PromptMRG |  AAAI 2024             |  0.160 |  0.213     |  0.229  |  0.211 |
> | REVTAF    |  ICCV 2025             |  0.176 |  0.286     |  0.282  |  0.273 |
> | our       |  -                     |  0.178 |  0.470     |  0.468  |  0.468 |
>
> **Q4: The theoretical analysis of CRS.**
>
> **R4:** We added Appendix A.18 (Line1449-1570) providing a theoretical analysis of CRS, showing it is grounded in input sensitivity and semantic consistency, formalized via information-theoretic (Shannon entropy) and representation-geometric (SBERT cosine similarity) measures. CRS thus reliably identifies high-leverage errors reflecting poor visual grounding.
>
> **Q5:Since LLMs lack true reasoning ability and tend to flag most regions as uncertain, relying on them to identify difficult cases fails to reduce physician workload.**
>
> **R5:** First, We fully agree that LLMs lack true reasoning and often produce unreliable uncertainty estimates—either overconfident or excessively uncertain—making them unsuitable for identifying difficult cases. Crucially, our method ​does not rely on the LLM’s self-assessed uncertainty, reasoning, or any notion of which concepts are “hard” for it to understand​. Instead, we only need to identify ​which parts of the generated report are at high risk of being inaccurate or inconsistent​, regardless of the cause.
>
> To this end, we propose the ​Concept Risk Score (CRS)​, which quantifies the instability of each anatomical concept by measuring output variations of our report generation model under input perturbations. CRS provides an ​objective, empirical, and model-intrinsic signal of potential error—without using any LLM confidence or self-awareness​. We then refer only the top-ranked high-risk concepts (by CRS) to physicians, significantly reducing annotation burden while directing human effort toward the most critical corrections.
>
> **Reference**
>
> [1] Ultrasound report generation with cross-modality feature alignment via unsupervised guidance, TMI, 2024
>
> [2] CheXbert: Combining Automatic Labelers and Expert Annotations for Accurate Radiology Report Labeling Using BERT, arXiv, 2020
>
> [3] Ultrasound Report Generation with Multimodal Large Language Models for Standardized Texts, arXiv, 2025
>
> [4] Preparing a collection of radiology examinations for distribution and retrieval, Journal of the American Medical Informatics Association, 2015

---

### Official Review · Reviewer_SJet · 2025-10-28

**Soundness:** 2
**Presentation:** 3
**Contribution:** 2
**Rating:** 6
**Confidence:** 2

**Summary:**

This paper proposes a new medical report generation framework, RCL-3 (Report Collaboration Level-3), which aims to improve report credibility and reduce physician burden through a bidirectional collaboration mechanism between physicians and AI. The core concept is that AI first generates a draft report, then automatically detects and labels high-risk misconceptions. Physicians then manually correct only these "Anatomy-Finding Concept Units" (AFCUs). Finally, the AI regenerates the report based on the feedback. The paper's main modules include: AFCU concept extraction, which extracts anatomy and finding concepts from reports using DeepSeek-V3; Information Bottleneck Clustering, which merges semantically similar concepts to reduce redundancy; Concept Risk Score (CRS), which calculates visual-semantic instability through occlusion perturbation to prioritize physician intervention; and the Holistic Semantic Match (HSM) metric, which measures the semantic consistency between model output and human annotations. The paper reports an average HSM improvement of approximately 9.13% on four organ datasets (thyroid, breast, liver, and ovary), and claims that the report quality can be significantly improved by modifying only one AFCU.

**Strengths:**

This study offers a new perspective by framing report generation as a human-machine collaborative problem rather than a purely generative task. The main strength of this paper lies in its directional insights and conceptually valuable attempt to shift medical report generation from one-off automation to collaborative AI design. The proposed AFCU granularity is intuitive and reasonable, providing a middle ground between sentence-level and token-level semantics. Multi-organ experiments also demonstrate some generalization capabilities.

**Weaknesses:**

1. Validation relies solely on machine-translated English datasets (from Chinese), risking distortion of nuanced clinical semantics; no tests on native English datasets (e.g., MIMIC-CXR) leave adaptability to English workflows uncertain.
2. The Concept Risk Score only detects high-risk concepts the model generates (e.g., incorrect nodules) but misses omitted critical concepts (e.g., unmentioned pleural effusions), limiting report safety.
3. The "single AFCU intervention" design lacks exploration of 2–3 high-risk AFCU corrections (for better efficiency-accuracy balance) and no discussion of low-risk error accumulation in practice.

**Questions:**

1. How is bidirectional collaboration actually realized? Did any physicians participate in providing real feedback, or is it fully simulated?
2. How does the proposed CRS compare quantitatively with standard uncertainty estimation methods (entropy, ensemble variance, semantic entropy, etc.)?
3. What criteria were used to set the Information Bottleneck coefficient β? How sensitive are the results to this hyperparameter?
4. How stable is the correlation between HSM and established metrics (BLEU, CheXbert, RadGraph-F1) across datasets or random seeds?
5. The paper assumes that “correcting one AFCU” suffices—how realistic is this in multi-pathology cases where multiple findings coexist?
6. Would the framework still function effectively if the base model (e.g., LLaVA, Gemini, or Qwen) were replaced?
7. Has the study quantified the physician interaction cost or time savings through any real or simulated user experiments?

---

> ### Author Response · Authors · 2025-11-24
> **Response to Reviewer SJet (Part 1/2)**
>
> Dear Reviewer SJet,
>
> We sincerely appreciate the reviewer for carefully reading our paper. We are encouraged by the positive comments.
>
> **Q1: Reliance on machine-translated data risks semantic distortion; lack of evaluation on native English datasets (e.g., MIMIC-CXR) undermines confidence in English clinical applicability.**
>
> **R1:** First, we have added evaluation results on the English-language IU-Xray dataset (Appendix A.15) and, in light of Reviewer TNks’s insightful comment—that clinical efficacy is more important than textual similarity—we specifically report clinical efficacy (CE) metrics (precision, recall, and F1). The results demonstrate that our method substantially improves the clinical efficacy of generated reports. Second, as stated in the paper (Line 374-375): “All data and experiments are in Chinese. English text in figures was translated from Chinese using Google Translate for readability.”
>
> |           | Year                   | METEOR |  Precision |  Recall |  F1    |
> |-|-|-|-|-|-|
> | R2Gen     |  ACL 2020              |  0.128 |  0.151     |  0.145  |  0.145 |
> | CVT2Dis   |  Artif.Intell.Med 2022 |  0.147 |  0.174     |  0.172  |  0.168 |
> | M2KT      |  MIA 2023              |  0.153 |  0.153     |  0.145  |  0.145 |
> | PromptMRG |  AAAI 2024             |  0.160 |  0.213     |  0.229  |  0.211 |
> | REVTAF    |  ICCV 2025             |  0.176 |  0.286     |  0.282  |  0.273 |
> | Ours       |  -                     |  0.178 |  0.470     |  0.468  |  0.468 |
>
> **Q2: Concept Risk Score misses omitted critical findings (e.g., pleural effusion), compromising report safety.**
>
> **R2**: We already included this issue in the Limitations section of our initial submission (Line 1305-1307). However, since the number of anatomical concepts is limited, we can compute CRS for all candidate anatomical concepts to avoid missing any, and we plan to explore this in future work.
>
> **Q3: Single-AFCU design omits exploration of multi-correction (2–3 high-risk AFCUs) and ignores low-risk error accumulation in practice.**
>
> **R3:** We have added new experiments, visualizations, and discussion (Sec. 4.7, Line 496-518) to investigate the impact of correcting 0–4 high-risk AFCUs on report refinement. The results show that iteratively correcting multiple high-risk AFCUs further improves model performance, though with diminishing returns. This indirectly validates that our CRS successfully identifies anatomical concepts with high error risk, and correcting them leads to performance gains.
>
> Importantly, CRS assesses ​*error risk*​, not certainty of error—meaning low-risk concepts are not necessarily incorrect. In practice, it may be acceptable to leave some low-risk concepts uncorrected, as their descriptions may differ from clinicians’ wording only in terminology while remaining semantically accurate and correct. Furthermore, the issue of error accumulation can be mitigated by refining multiple AFCUs, which enhances the overall quality of the generated reports.
>
> **Q4: How is bidirectional collaboration implemented—through real physician feedback or fully simulated?**
>
> **R4:**  We have added a detailed description in the Method section (Line 185-188): (1) A fine-tuned VLM generates an initial report; (2) CRS identifies the anatomical concept with the highest error risk, and a physician provides the corrected description; (3) The VLM incorporates this feedback to produce the final revised report. Moreover, our method is entirely simulation-based.
>
> **Q5: How does CRS quantitatively compare to standard uncertainty methods ？**
>
> **R5:** Standard uncertainty estimation methods (e.g., entropy, ensemble variance) typically operate at the sentence level, which does not align with our task’s requirement to assess uncertainty at the anatomical concept level. To ensure a fair comparison, we adapt all baseline methods to compute uncertainty scores for the textual span corresponding to each anatomical concept—exactly as done in our CRS. We have clarified this implementation detail in Line 469-470 of the paper.
>
> **Q6：What criteria were used to set the Information Bottleneck coefficient β? How sensitive are the results to this hyperparameter?**
>
> **R6:** First, the Information Bottleneck coefficient β controls the perceptual similarity threshold for grouping finding concepts and is selected and adjusted as needed by researchers with three years of medical background. Consequently, the results are indeed closely tied to this hyperparameter; however, because medical experts are involved in the process, the resulting concepts are clinically discriminative and practically usable.

---

> > ### Author Response · Authors · 2025-11-24
> > **Response to Reviewer SJet (Part 2/2)**
> >
> > **Q7: How stable is the correlation between HSM and established metrics (BLEU, CheXbert, RadGraph-F1) across datasets or random seeds?**
> >
> > **R7:** We have added a new analysis in the Appendix A.13 (Line 1329-1339)  to address this. Our results show that the correlation between HSM and BLEU is strong for organs with standardized reporting (e.g., Liver, Mammary; ​R​>0.83) but weaker for more complex cases like Ovary (R=0.55), where BLEU fails to capture clinically critical errors. This indicates that HSM provides a more consistent and clinically grounded evaluation across diverse anatomical contexts, whereas traditional metrics can be unstable or misleading depending on the dataset’s semantic complexity.
> >
> > | Organ    | Sample Size | HSM Mean | BLEU1 Mean | BLEU4 Mean | HSM–BLEU1 |          | HSM–BLEU4 |          |
> > |-|-|-|-|-|-|-|-|-|
> > |          |             |          |            |            | R         | p-value  | R         | p-value  |
> > | Liver    | 279         | 0.8449   | 0.8854     | 0.8379     | 0.8467    | p<0.001  | 0.8735    | p<0.001  |
> > | Mammary  | 703         | 0.7064   | 0.7228     | 0.6557     | 0.8335    | p<0.001  | 0.8896    | p<0.001  |
> > | Ovary    | 169         | 0.4673   | 0.5901     | 0.4415     | 0.5480    | p<0.001  | 0.6979    | p<0.001  |
> >
> > **Q8: The paper assumes that “correcting one AFCU” suffices—how realistic is this in multi-pathology cases where multiple findings coexist?**
> >
> > **R8:** We have added new experiments, visualizations, and discussion (Sec. 4.7, Line 498-518) to explore the impact of correcting 0–4 high-risk AFCUs on report refinement. The results show that iteratively correcting multiple high-risk AFCUs further improves model performance. Additionally, as this is our initial exploration and no established baselines exist for such multi-correction interaction, we chose to start by studying the single-correction collaborative setting. Future work will focus on enhancing the performance and efficiency of multi-correction collaborative paradigms.
> >
> > **Q9: Would the framework still function effectively if the base model (e.g., LLaVA, Gemini, or Qwen) were replaced?**
> >
> > **R9:** Yes, although the main paper adopts Qwen2.5-VL as the primary VLM, we discuss in Appendix A.10 (Line 1224-1241) that combining our method with MedGemma-4B still yields strong performance.
> >
> > | Model              | BLEU-1  | BLEU-4  | METEOR  | ROUGE-L  | Precision  | Recall  | F1 Score  | AIOU    | FSS     | HSM    |
> > |-|-|-|-|-|-|-|-|-|-|-|
> > | Qwen2.5-VL-3B+SFT  | 0.6220  | 0.4367  | 0.3284  | 0.6042   | 0.8541     | 0.8215  | 0.8260    | 0.7281  | 0.5622  | 0.6351 |
> > | Medgemma-4b+SFT    | 0.6248  | 0.4350  | 0.3318  | 0.6039   | 0.8635     | 0.8219  | 0.8311    | 0.7378  | 0.5712  | 0.6445 |
> > | Medgemma-4b+Ours   | 0.6642  | 0.4609  | 0.3558  | 0.6107   | 0.8879     | 0.8676  | 0.8689    | 0.7877  | 0.5916  | 0.6777 |
> >
> > **Q10: Has the study quantified the physician interaction cost or time savings through any real or simulated user experiments?**
> >
> > **R10:** We have added a new experiment that quantifies the time savings through a human-in-the-loop evaluation. In Appendix A.16 (Line 1388-1403), we report results from an experiment using 15 samples from the Mammary dataset, comparing four annotation settings. Our method—AI-generated reports with CRS-based checking followed by physician editing of one AFCU—reduced the average annotation time by 14.34 seconds compared to conventional AI-assisted editing without CRS, demonstrating a measurable reduction in physician workload.
> >
> > | Human-Generated  | AI-Generated  | Human Checking  | CRS  | Human Editing  | Time (s) |
> > |-|-|-|-|-|-|
> > | √                |               |                 |      |                | 205.95   |
> > |                  | √             |                 |      |                | 15.16    |
> > |                  | √             | √               |      | √              | 46.10    |
> > |                  | √             |                 | √    | √              | 31.76    |

---

### Official Review · Reviewer_tMhq · 2025-11-01

**Soundness:** 4
**Presentation:** 4
**Contribution:** 4
**Rating:** 8
**Confidence:** 5

**Summary:**

The paper propose the Anatomy-Finding Concept Unit (AFCU), a minimal, clinically grounded semantic statement satisfying three key principles: atomicity, lack of ambiguity, and anatomical anchoring. To extract AFCU, we use a Large Language Model (LLM) guided by pre-
defined clinical templates followed by information bottleneck clustering to group lexically diverse but semantically equivalent anatomical concepts. To prioritize physician intervention, the paper introduce the Concept Risk Score (CRS), quantifying behavioral inconsistency and semantic instability. Experiments show the proposed framework improves semantic quality by 9.13% HSM across four organs.

**Strengths:**

1. Echocardiography report generation with a human-in-the-loop is both compelling and clinically important.

2. The framework is original; the proposed modules—Anatomy–Finding Concept Unit (AFCU), concept risk scoring, and holistic semantic matching—are well-motivated and valuable to the community.

3. Extensive experiments demonstrate the method’s effectiveness, and the paper includes sufficient implementation details to support reimplementation.

**Weaknesses:**

1. The paper states (Line 259) that no retraining is required and that the model can follow intervention prompts to refine its outputs. Please clarify whether this capability is inherited from the base VLM (e.g., Qwen) or emerges from your framework. If it mainly comes from the base model, consider adding a small supervised fine-tuning (SFT) set with refinement prompts to improve controllability and consistency. A comparison among (i) base Qwen with prompts, (ii) your method without retraining, and (iii) your method with light SFT would isolate the contribution of each component.

2. The table is difficult to interpret because the definitions of “phrase-level,” “sentence-level,” and “report-level” SFT are not specified (task formulation, supervision granularity, data size). The current takeaway appears to be that AFCU-derived concepts outperform GPT-generated concepts—which is plausible—but report-level SFT yields the strongest results overall. Please explain the rationale behind these trends, control for data volume, and state what the table is intended to demonstrate (e.g., the value of concept granularity versus holistic supervision).

**Questions:**

1.  Qwen2.5-VL-3B outperforms the 7B model on many metrics. What factors drive this (e.g., instruction tuning mismatch, optimization stability, decoding settings, or domain overfitting)?

2. Would you consider releasing code, prompts, and (where permissible) data specifications to facilitate replication and downstream use?

---

> ### Author Response · Authors · 2025-11-24
> **Response to Reviewer tMhq**
>
> Dear Reviewer tMhq,
>
> At the beginning, we want to express our sincere gratitude to the reviewer for the positive feedbacks. We appreciate your recognition of the novelty and effectiveness of our method.
>
> **Q1: Is the prompt-following ability from the base VLM or your framework?**
>
> **R1:** We have added a new set of ablation experiments (see Appendix A.14, Line 1342-1349), which demonstrate that the prompt-following ability comes not from the base VLM, but from our framework—specifically, Concept Instruction Tuning. As shown in Table, the base QwenVL-3B with intervention prompts (“Base + Prompt”) performs poorly, whereas our fine-tuned model achieves strong results, confirming that this capability is acquired through our training strategy, not inherent to the base model.
>
> |                       | BLEU-1 | BLEU-4  |  METEOR |  ROUGE-L |
> |-|-|-|-|-|
> | Base + Prompt         | 0.1043 | 0.0301  | 0.1327  | 0.3478   |
> | Concept Instruction Tuning + Prompt|0.6951|0.5931|0.4342|0.7078|
> | Concept Instruction Tuning + Prompt+ CRS|0.8064 | 0.7080  | 0.4713  | 0.7883   |
> | Upper Bound           | 0.8469 | 0.7666  | 0.5095  | 0.8300   |
>
> **Q2: Unclear SFT definitions; explain why report-level SFT works best, control for data volume, and clarify the table’s purpose**
>
> **R2:** We have added a detailed discussion of this part in the Appendix A.19, Line 1575-1605. All SFT variants use the same dataset size. Report-level SFT achieves the best performance because it uses the full ground-truth report as input—serving only as a theoretical upper bound, not a practical human–AI collaboration method. The table compares different supervision granularities to justify AFCU as the optimal unit for interaction: unlike overly coarse (report/sentence-level) or semantically incomplete (phrase-level) inputs, and ambiguous GPT concepts, AFCU provides fine-grained, anatomically anchored, and clinically actionable correction units.
>
> | Type                     | BLEU-4  | ROUGE-L | HSM     |
> |-|-|-|-|
> | SFT                      | 0.6341  | 0.7277  | 0.6475  |
> | + Phrase-level           | 0.5662  | 0.6775  |  0.5984 |
> | + Sentence-level         | 0.6481  | 0.7380  | 0.6280  |
> | + Report-level           | 0.8669  | 0.8999  | 0.8681  |
> | Concept-level (GPT) SFT  | 0.5653  | 0.6434  | 0.5602  |
> | + Concept-level (GPT)    | 0.5131  | 0.6370  | 0.5470  |
> | Concept-level (AFCU) SFT | 0.6230  | 0.6985  | 0.6282  |
> | + Concept-level (AFCU)   | 0.6604  | 0.7413  | 0.7015  |
>
> **Q3: Why does Qwen2.5-VL-3B outperform the 7B model?**
>
> **R3:** Indeed, on certain smaller datasets, the Qwen2.5-VL-3B model outperforms the 7B variant—likely because the larger model’s higher capacity leads to overfitting when training data is limited. We have added a clarification to this effect in the revised manuscript (Line 421-424). In future work, we plan to incorporate larger-scale instruction-tuning datasets to better harness the potential of bigger models.
>
> **Q4: Will you release code, prompts, and data specs for reproducibility?**
>
> **R4:** Yes, we will release the code, prompts, and other relevant materials upon acceptance. Links to the public datasets used in this paper will also be provided.

---

### Author Response · Authors · 2025-11-24
**Response to All Reviewers**

Dear Reviewers,

We sincerely thank you for your valuable feedback and positive recognition. Our work received favorable comments on its soundness (`tMhq, yPVd, kJoU`), contribution (`tMhq, yPVd`), and clear, easy-to-follow presentation (`tMhq, SJet, TNks`). The reviewers also highlighted the clinical relevance and practical significance of our research problem (`tMhq, SJet`), as well as the originality and well-motivated design of our approach (`tMhq, SJet, TNks`).

We have carefully addressed all comments with targeted revisions and provided point-by-point responses. All changes in the revised manuscript are highlighted in blue for ease of review.

Should you have any further questions, we would be happy to clarify.


Best regards,

Authors

---

### Author Response · Authors · 2025-12-03
**Summary of Revisions and Responses for Submission 4639 (Part 1/2)**

Dear PCs, SACs, ACs, and Reviewers,

We deeply appreciate your time and effort dedicated to the review process. To support the newly assigned Area Chair and help streamline their evaluation process, we provide below a concise summary of the key points from the reviews and our responses.

**1. Reviewer `yPVd`**

Reviewer `yPVd` upgraded our score from 2 to 4 on November 26, 2025, at 18:42:18. Their remaining concerns primarily focus on (i) the collaborative interaction mechanism, (ii) Generalizability of the method and (iii) the clarity of the paper.

`Regarding (i)`, we have added Appendix A.21, which demonstrates that the core of our collaborative interaction mechanism lies in its fine-grained, strongly anchored binding of textual descriptions to specific anatomical structures—outperforming existing correction approaches.

`In terms of (ii)`, we demonstrate the effectiveness of our approach across four anatomical organs using Qwen-VL-3B, Qwen-VL-7B, and MedGemma-4B, as shown in Table 2 (Lines 380–416) and the newly added experimental results (Lines 1242–1250), highlighting the broad applicability of our method.

`For (iii)`, we have added a clear description of the human–AI collaboration workflow in Lines 185–188 of the Method section and provided a fully English version of this workflow in Appendix A.17. Moreover, we have included extensive supplementary materials (Appendices A.13–A.21) to clarify any potentially ambiguous points. Notably, reviewers tMhq, SJet, and TNks all explicitly stated in their initial reviews that our paper features a clear and easy-to-follow presentation.

**2. Reviewer `kJoU`**

In their response dated November 25, 2025, at 18:26:40, Reviewer `kJoU` acknowledged parts of our rebuttal and recognized that the newly added content adds value to the paper. Regarding their lingering concerns about Q6 and Q9, `kJoU suggested conducting a new experiment` to verify whether our AFCU-based correction outperforms Radialog specifically in the correction phase, noting that “this would also be a relevant result” if proven true. `We strictly followed this suggestion and added the new experiment` in Appendix A.21, which clearly shows that our AFCU method surpasses Radialog in correction effectiveness—thus resolving this concern.

Regarding kJoU’s other concern—that `our contribution is still overclaimed, and that our work could only be reduced to uncertainty estimation unless we demonstrated AFCU’s superiority in correction`—`we have now provided exactly that evidence via Appendix A.21`. As kJoU themselves noted, such a result would constitute a meaningful contribution, which our experiments now substantiate.

More importantly, we explicitly define three Radiology Collaboration Levels (RCL) in Figure 1 and Lines 48–57 of the Introduction: Radialog falls under RCL-1 (one-way assistance), whereas our work presents the first RCL-3 framework (risk-aware, proactive collaboration). To the best of our knowledge, no prior work has explored this human–AI interaction paradigm. Therefore, our concept-level modeling and formalization of this novel framework constitute a unique contribution. Furthermore, we maintain that clearly articulating multiple contributions does not equate to overclaiming; rather, it reflects our progressive understanding of the problem and the corresponding solution design. We hope that summarizing these key methodological insights will offer meaningful inspiration to the community.

---

> ### Author Response · Authors · 2025-12-03
> **Summary of Revisions and Responses for Submission 4639 (Part 2/2)**
>
> **3. Reviewer `TNks`**
>
> Reviewer `TNks` raised five specific issues: (i) improving readability affected by Chinese content; (ii) adding Clinical Efficacy (CE) metrics; (iii) including comparisons with recent methods; (iv) providing a theoretical analysis of CRS; (v) clarifying that relying on LLMs’ self-assessed uncertainty fails to reduce physician workload.We addressed `(i)`by adding Appendix A.17 (“Language Presentation and Accessibility”) with fully English versions of all figures.`For (ii) and (iii)`, we added Appendix A.15 (“Enhancing Clinical Efficacy on the English IU-Xray Dataset”), which includes CE metrics and comparisons with state-of-the-art methods (e.g., ICCV 2025, AAAI 2024). `For (iv)`, we added Appendix A.18 (“Theoretical Analysis of the Concept Risk Score”). `For (v)`, we clarified that our method does not rely on the LLM’s self-assessed uncertainty or reasoning capabilities; instead, we identify report segments at high risk of inaccuracy or inconsistency—regardless of the cause—using an objective, perturbation-based signal.
>
> **4. Reviewer `tMhq` and `SJet`**
>
> We have also fully addressed all questions from reviewers `tMhq` and `SJet`. Notably, our new Section 4.7 (“Impact of Multiple CRS Concept Injection”) demonstrates that our method remains effective even when correcting multiple localized errors simultaneously, highlighting its strong potential as a new paradigm for human–AI collaborative report generation.
>
> Finally, we quote two key endorsements from `tMhq` and `SJet` to underscore the `significance of our proposed paradigm`:`“report generation with a human-in-the-loop is both compelling and clinically important.”`, `“This study offers a new perspective by framing report generation as a human-machine collaborative problem rather than a purely generative task.”` Unlike RCL-1 and RCL-2 approaches—which require physicians to comprehensively review both the AI-generated report and the full medical image—our RCL-3 framework enables physicians to focus only on localized image regions and associated text errors, guided by risk signals and concept-level interaction instructions. `This significantly reduces clinical workload and establishes a novel, impactful direction for medical report generation research`.
>
> We believe these revisions have substantially strengthened the paper and hope you find our work suitable for acceptance.
>
> Sincerely,
>
> Authors of Submission 4639
>
> Submission ID: 4639

---

### Meta-Review · Area_Chair_bz3w · 2026-01-06

**Summary:**

This paper proposes concept-level human–AI collaboration for radiology report generation.  The paper introduces new components (AFCU, CRS, HSM) and the authors added a substantial rebuttal with new ablations, theory, CE metrics, English-only figures, and resolved several other concerns.
The authors made a significant effort in the rebuttal and substantially improved the manuscript with many useful analyses. Still, the key experimental isolation (correction-only ablation across competitive baselines) and safety/omission demonstrations are missing or not convincing. Because those gaps are central to the main claim, I recommend rejection at this round but encourage a strong revision addressing the reviewers' comments and to do the resubmission elsewhere.

**Reviewer Concerns:**

Core experimental isolation of the correction step.
Multiple reviewers requested the specific ablation the authors did not convincingly perform. The authors present some prompting experiments, but reviewers argue comparisons are not the direct, fair ablation needed.  One reviewer insists the paper still lacks an experiment where other methods are given the same correction capability. This is the single clearest outstanding empirical gap.

Authors state CRS can be computed over all candidate anatomical concepts, but do not convincingly demonstrate detection of omitted critical findings in practice or show recall/coverage statistics for omission detection.
The interaction is still simulated. No real physician-in-the-loop study is provided; reviewers called this out as limiting external validity.

**Reviewer Scores:**

Only one reviewer seems to have changed their score. The rest was hovering around their low score to begin with.

---

### Decision · Program_Chairs · 2026-01-26

Reject